# Characterization of a New Member of Alphacoronavirus with Unique Genomic Features in Rhinolophus Bats

**DOI:** 10.3390/v11040379

**Published:** 2019-04-24

**Authors:** Ning Wang, Chuming Luo, Haizhou Liu, Xinglou Yang, Ben Hu, Wei Zhang, Bei Li, Yan Zhu, Guangjian Zhu, Xurui Shen, Cheng Peng, Zhengli Shi

**Affiliations:** 1CAS Key Laboratory of Special Pathogens and Biosafety, Wuhan Institute of Virology, Chinese Academy of Sciences, Wuhan 430071, China; wuzhongxiaoliu@163.com (N.W.); chine.lcm@163.com (C.L.); yangxl@wh.iov.cn (X.Y.); ouihaagendazs@hotmail.com (B.H.); happyzhang_2007@163.com (W.Z.); shenqi0622@126.com (B.L.); zhuyan@wh.iov.cn (Y.Z.); pukulovesu@163.com (X.S.); pengcheng@wh.iov.cn (C.P.); 2College of Life Science, University of Chinese Academy of Sciences, Beijing 100864, China; 3Center for Emerging Infectious Disease, Wuhan Institute of Virology, Chinese Academy of Sciences, Wuhan 430071, China; liuhz@wh.iov.cn; 4EcoHealth Alliance, New York, NY 10001, USA; zhugj609@126.com

**Keywords:** coronavirus, alphacoronavirus, *Rhinolophus* bat, unique genes

## Abstract

Bats have been identified as a natural reservoir of a variety of coronaviruses (CoVs). Several of them have caused diseases in humans and domestic animals by interspecies transmission. Considering the diversity of bat coronaviruses, bat species and populations, we expect to discover more bat CoVs through virus surveillance. In this study, we described a new member of alphaCoV (BtCoV/Rh/YN2012) in bats with unique genome features. Unique accessory genes, *ORF4a* and *ORF4b* were found between the spike gene and the envelope gene, while *ORF8* gene was found downstream of the nucleocapsid gene. All the putative genes were further confirmed by reverse-transcription analyses. One unique gene at the 3’ end of the BtCoV/Rh/YN2012 genome, *ORF9*, exhibits ~30% amino acid identity to *ORF7a* of the SARS-related coronavirus. Functional analysis showed ORF4a protein can activate IFN-β production, whereas ORF3a can regulate NF-κB production. We also screened the spike-mediated virus entry using the spike-pseudotyped retroviruses system, although failed to find any fully permissive cells. Our results expand the knowledge on the genetic diversity of bat coronaviruses. Continuous screening of bat viruses will help us further understand the important role played by bats in coronavirus evolution and transmission.

## 1. Introduction

Members of the *Coronaviridae* family are enveloped, non-segmented, positive-strand RNA viruses with genome sizes ranging from 26–32 kb [1]. These viruses are classified into two subfamilies: *Letovirinae*, which contains the only genus: *Alphaletovirus*; and *Orthocoronavirinae* (*CoV*), which consists of alpha, beta, gamma, and deltacoronaviruses (CoVs) [2,3]. Alpha and betacoronaviruses mainly infect mammals and cause human and animal diseases. Gamma- and delta-CoVs mainly infect birds, but some can also infect mammals [4,5]. Six human CoVs (HCoVs) are known to cause human diseases. HCoV-HKU1, HCoV-OC43, HCoV-229E, and HCoV-NL63 commonly cause mild respiratory illness or asymptomatic infection; however, severe acute respiratory syndrome coronavirus (SARS-CoV) and Middle East respiratory syndrome coronavirus (MERS-CoV) have caused severe disease with a 10% or 35% mortality, respectively [6]. CoVs infection in domestic animals can also cause great economic losses, such as transmissible gastroenteritis virus, porcine epidemic diarrhea virus, and HKU2-related CoV in pigs [7,8,9,10].

Prior to the global SARS outbreak in 2002 to 2003, only 10 CoVs were reported. Since the outbreak, numerous CoVs have been discovered in animals, particularly, in bats [6]. According to a recent report by the International Committee of Viruses on Taxonomy (ICTV), at least 17 out of 29 assigned alpha and beta-CoV species were identified from 11 out of 18 bat families [2,3]. Phylogenetic analysis suggested that bats are major hosts for alpha- and beta-CoVs. Recombination of different CoVs occurred in bats, as previously reported. Bats play an important role in CoV evolution [4].

Rhinolophus bats are widespread in China. At least 4 CoV species with high genetic diversity have been found in members of this family [11]. Among these viruses, bat SARS-related coronaviruses (SARSr-CoVs) have been proved to be able to infect animal and human cells by using the same receptor as SARS-CoV [12,13,14]. Recently, a new porcine disease was confirmed to be caused by BatCoV HKU2-related virus in Guangdong Province, China [8,9,15]. These findings indicate that these bat species play important roles in CoV evolution and transmission.

Here, we report a novel species of alpha-CoV discovered in *Rhinolophus* bats in China, their unique genomic structures and a preliminary functional assessment of accessory genes, as well as this virus’ infectivity in different cells.

## 2. Materials and Methods

### 2.1. Ethics Statement

All sampling procedures were performed by veterinarians, with approval from Animal Ethics Committee of the Wuhan Institute of Virology (WIVH5210201). The study was conducted in accordance with the Guide for the Care and Use of Wild Mammals in Research of the People’s Republic of China.

### 2.2. Sampling

Bat fecal swab and pellet samples were collected from November 2004 to November 2014 in different seasons in Southern China, as described previously [16].

### 2.3. RNA Extraction, PCR Screening and Sequencing

Viral RNA was extracted from 200 μL of fecal swab or pellet samples using the High Pure Viral RNA Kit (Roche Diagnostics GmbH, Mannheim, Germany) as per the manufacturer’s instructions. RNA was eluted in 50 μL of elution buffer, aliquoted, and stored at –80 °C. One-step hemi-nested reverse-transcription (RT-) PCR (Invitrogen, San Diego, CA, USA) was employed to detect coronavirus, as previously described [17,18].

To confirm the bat species of an individual sample, we PCR amplified the cytochrome b (Cytob) and/or NADH dehydrogenase subunit 1 (ND1) gene using DNA extracted from the feces or swabs [19,20]. The gene sequences were assembled excluding the primer sequences. BLASTN was used to identify host species based on the most closely related sequences with the highest query coverage and a minimum identity of 95%.

### 2.4. Sequencing of Full-Length Genomes

Full genomic sequences were determined by one-step PCR (Invitrogen, San Diego, CA, USA) amplification with degenerate primers (Appendix A) designed on the basis of multiple alignments of available alpha-CoV sequences deposited in GenBank or amplified with SuperScript IV Reverse Transcriptase (Invitrogen) and Expand Long Template PCR System (Roche Diagnostics GmbH, Mannheim, Germany) with specific primers (primer sequences are available upon request). Sequences of the 5’ and 3’ genomic ends were obtained by 5’ and 3’ rapid amplification of cDNA ends (SMARTer RACE 5’/3’ Kit; Clontech, Mountain View, CA, USA), respectively. PCR products were gel-purified and subjected directly to sequencing. PCR products over 5kb were subjected to deep sequencing using Hiseq2500 system. For some fragments, the PCR products were cloned into the pGEM-T Easy Vector (Promega, Madison, WI, USA) for sequencing. At least five independent clones were sequenced to obtain a consensus sequence.

### 2.5. Genome Analysis

The Next Generation Sequencing (NGS) data were filtered and mapped to the reference sequence of BatCoV HKU10 (GenBank accession number NC_018871) using Geneious 7.1.8 [21]. Genomes were preliminarily assembled using DNAStar lasergene V7 (DNAStar, Madison, WI, USA). Putative open reading frames (ORFs) were predicted using NCBI’s ORF finder (https://www.ncbi.nlm.nih.gov/orffinder/) with a minimal ORF length of 150 nt, followed by manual inspection. The sequences of the 5’ untranslated region (5’-UTR) and 3’-UTR were defined, and the leader sequence, the leader and body transcriptional regulatory sequence (TRS) were identified as previously described [22]. The cleavage of the 16 nonstructural proteins coded by ORF1ab was determined by alignment of aa sequences of other CoVs and the recognition pattern of the 3C-like proteinase and papain-like proteinase. Phylogenetic trees based on nt or aa sequences were constructed using the maximum likelihood algorithm with bootstrap values determined by 1000 replicates in the MEGA 6 software package [23]. Full-length genome sequences obtained in this study were aligned with those of previously reported alpha-CoVs using MUSCLE [24]. The aligned sequences were scanned for recombination events by using Recombination Detection Program [25]. Potential recombination events as suggested by strong *p*-values (<10^–20^) were confirmed using similarity plot and bootscan analyses implemented in Simplot 3.5.1 [26]. The number of synonymous substitutions per synonymous site, Ks, and the number of nonsynonymous substitutions per nonsynonymous site, Ka, for each coding region were calculated using the Ka/Ks calculation tool of the Norwegian Bioinformatics Platform (http://services.cbu.uib.no/tools/kaks) with default parameters [27]. The protein homology detection was analyzed using HHpred (https://toolkit.tuebingen.mpg.de/#/tools/hhpred) with default parameters [28].

### 2.6. Transcriptional Analysis of Subgenomic mRNA

A set of nested RT-PCRs was employed to determine the presence of viral subgenomic mRNAs in the CoV-positive samples [29]. Forward primers were designed targeting the leader sequence at the 5’-end of the complete genome, while reverse primers were designed within the ORFs. Specific and suspected amplicons of expected sizes were purified and then cloned into the pGEM-T Easy vector for sequencing.

### 2.7. Cell Lines, Gene Cloning, and Expression

Bat primary or immortalized cells (*Rhinolophus sinicus* kidney immortalized cells, RsKT; *Rhinolophus sinicus* Lung primary cells, RsLu4323; *Rhinolophus sinicus* brain immortalized cells, RsBrT; *Rhinolophus affinis* kidney primary cells, RaK4324; *Rousettus leschenaultii* Kidney immortalized cells, RlKT; *Hipposideros pratti* lung immortalized cells, HpLuT) generated in our laboratory were all cultured in DMEM/F12 with 15% FBS. *Pteropus alecto* kidney cells (Paki) was maintained in DMEM/F12 supplemented with 10% FBS. Other cells were maintained according to the recommendations of American Type Culture Collection (ATCC, www.atcc.org).

The putative accessory genes of the newly detected virus were generated by RT-PCR from viral RNA extracted from fecal samples, as described previously [30]. The influenza virus NS1 plasmid was generated in our lab [31]. The human bocavirus (HBoV) VP2 plasmid was kindly provided by prof. Hanzhong Wang of the Wuhan Institute of Virology, Chinese Academy of Sciences. SARS-CoV ORF7a was synthesized by Sangon Biotech. The transfections were performed with Lipofectamine 3000 Reagent (Life Technologies). Expression of these accessory genes were analyzed by Western blotting using an mAb (Roche Diagnostics GmbH, Mannheim, Germany) against the HA tag.

### 2.8. Virus Isolation

The virus isolation was performed as previously described [12]. Briefly, fecal supernatant was acquired via gradient centrifugation and then added to Vero E6 cells, 1:10 diluted in DMEM. After incubation at 37 ℃ for 1 h the inoculum was replaced by fresh DMEM containing 2% FBS and the antibiotic-antimycotic (Gibco, Grand Island, NY, USA). Three blind passages were carried out. Cells were checked daily for cytopathic effect. Both culture supernatant and cell pellet were examined for CoV by RT-PCR [17].

### 2.9. Apoptosis Analysis

Apoptosis was analyzed as previously described [18]. Briefly, 293T cells in 12-well plates were transfected with 3 μg of expression plasmid or empty vector, and the cells were collected 24 h post transfection. Apoptosis was detected by flow cytometry using by the Annexin V-FITC/PI Apoptosis Detection Kit (YEASEN, Shanghai, China) following the manufacturer’s instructions. Annexin-V-positive and PI-negative cells were considered to be in the early apoptotic phase and those stained for both Annexin V and PI were deemed to undergo late apoptosis or necrosis. All experiments were repeated three times. Student’s *t*-test was used to evaluate the data, with *p* < 0.05 considered significant.

### 2.10. Dual Luciferase Reporter Assays

HEK 293T cells were seeded in 24-well plates and then co-transfected with reporter plasmids (pRL-TK and pIFN-βIFN- or pNF-κB-Luc) [30], as well as plasmids expressing accessory genes, empty vector plasmid pcAGGS, influenza virus NS1 [32], SARS-CoV ORF7a [33], or HBoV VP2 [34]. At 24 h post transfection, cells were treated with Sendai virus (SeV) (100 hemagglutinin units [HAU]/mL) or human tumor necrosis factor alpha (TNF-α; R&D system) for 6 h to activate IFNβ or NF-κB, respectively. Cell lysates were prepared, and luciferase activity was measured using the dual-luciferase assay kit (Promega, Madison, WI, USA) according to the manufacturer’s instructions.

### 2.11. BtCoV/Rh/YN2012 Spike-Mediated Pseudoviruses Cell Tropism Screening

Retroviruses pseudotyped with BtCoV/Rh/YN2012 RsYN1, RsYN3, RaGD, or MERS-CoV spike, or no spike (mock) were used to infect human, bat or other mammalian cells in 96-well plates. The pseudovirus particles were confirmed with Western blotting and negative-staining electromicroscopy. The production process, measurements of infection and luciferase activity were conducted, as described previously [35,36].

### 2.12. Nucleotide Sequence Accession Numbers

The complete genome nucleotide sequences of BtCoV/Rh/YN2012 strains RsYN1, RsYN2, RsYN3, and RaGD obtained in this study have been submitted to the GenBank under MG916901 to MG916904.

## 3. Results

### 3.1. CoVs Detected in Rhinolophus Bats

The surveillance was performed between November 2004 to November 2014 in 19 provinces of China. In total, 2061 fecal samples were collected from at least 12 Rhinolophus bat species (Figure 1A). CoVs were detected in 209 of these samples (Figure 1B and Table 1). Partial *RdRp* sequences suggested the presence of at least 8 different CoVs. Five of these viruses are related to known species: *Mi-BatCoV 1* (>94% nt identity), *Mi-BatCoV HKU8* [37] (>93% nt identity), *BtRf-AlphaCoV/HuB2013* [11] (>99% nt identity), *SARSr-CoV* [38] (>89% nt identity), and *HKU2-related CoV* [39] (>85% nt identity). While the other three CoV sequences showed less than 83% nt identity to known CoV species. These three viruses should represent novel CoV species. Virus isolation was performed as previously described [12], but was not successful.

### 3.2. Genomic Characterization of a Novel Alpha-CoV (BtCoV/Rh/YN2012)

We next characterized a novel alpha-CoV, BtCoV/Rh/YN2012. It was detected in 3 *R.affinis* and 6 *R.sinicus*, respectively. Based on the sequences, we defined three genotypes, which represented by RsYN1, RsYN3, and RaGD, respectively. Strain RsYN2 was classified into the RsYN3 genotype. Four full-length genomes were obtained. Three of them were from *R.sinicus* (Strain RsYN1, RsYN2, and RsYN3), while the other one was from *R.affinis* (Strain RaGD). The sizes of these 4 genomes are between 28,715 to 29,102, with G+C contents between 39.0% to 41.3%. The genomes exhibit similar structures and transcription regulatory sequences (TRS) that are identical to those of other alpha-CoVs (Figure 2 and Table 2). Exceptions including three additional ORFs (ORF3b, ORF4a and ORF4b) were observed. All the 4 strains have ORF4a & ORF4b, while only strain RsYN1 has ORF3b.

The replicase gene, *ORF1ab*, occupies ~20.4 kb of the genome. The replicase gene, ORF1ab, occupies ~20.4 kb of the genome. It encodes polyproteins 1a and 1ab, which could be cleaved into 16 non-structural proteins (Nsp1–Nsp16). The 3’-end of the cleavage sites recognized by 3C-like proteinase (Nsp4-Nsp10, Nsp12-Nsp16) and papain-like proteinase (Nsp1–Nsp3) were confirmed. The proteins including Nsp3 (papain-like 2 proteas, PL2pro), Nsp5 (chymotrypsin-like protease, 3CLpro), Nsp12 (RdRp), Nsp13 (helicase), and other proteins of unknown function (Table 3). The 7 concatenated domains of polyprotein 1 shared <90% aa sequence identity with those of other known alpha-CoVs (Table 2), suggesting that these viruses represent a novel CoV species within the alpha-CoV. The closest assigned CoV species to BtCoV/Rh/YN2012 are BtCoV-HKU10 and BtRf-AlphaCoV/Hub2013. The three strains from Yunnan Province were clustered into two genotypes (83% genome identity) correlated to their sampling location. The third genotype represented by strain RaGD was isolated to strains found in Yunnan (<75.4% genome identity).

We then examined the individual genes (Table 2). All of the genes showed low aa sequence identity to known CoVs. The four strains of BtCoV/Rh/YN2012 showed genetic diversity among all different genes except ORF1ab (>83.7% aa identity). Notably, the spike proteins are highly divergent among these strains. Other structure proteins (E, M, and N) are more conserved than the spike and other accessory proteins. Comparing the accessory genes among these four strains revealed that the strains of the same genotype shared a 100% identical ORF3a. However, the proteins encoded by ORF3as were highly divergent among different genotypes (<65% aa identity). The putative accessory genes were also BLASTed against GenBank records. Most accessory genes have no homologues in GenBank-database, except for ORF3a (52.0–55.5% aa identity with BatCoV HKU10 ORF3) and ORF9 (28.1–32.0% aa identity with SARSr-CoV ORF7a). We analyzed the protein homology with HHpred software. The results showed that ORF9s and SARS-CoV OR7a are homologues (possibility: 100%, E value <10^−48^). We further screened the genomes for potential recombination evidence. No significant recombination breakpoint was detected by bootscan analysis.

### 3.3. Subgenomic Structures and Accessory Genes of BtCoV/Rh/YN2012

To confirm the presence of subgenomic RNA, we designed a set of primers targeting all the predicted ORFs as described. The amplicons were firstly confirmed via agarose-gel electrophoresis and then sequencing (Figure 3 and Table 2). The sequences showed that all the ORFs, except ORF4b, had preceding TRS. Hence, the ORF4b may be translated from bicistronic mRNAs. In RsYN1, an additional subgenomic RNA starting inside the ORF3a was found through sequencing, which led to a unique ORF3b.

### 3.4. Phylogenetic Analysis

Phylogenetic trees were constructed using the aa sequences of RdRp and S of BtCoV/Rh/YN2012 and other representative CoVs (Figure 4). In both trees, all BtCoV/Rh/YN2012 were clustered together and formed a distinct lineage to other known coronavirus species. Two distinct sublineages were observed within BtCoV/Rh/YN2012. One was from Ra sampled in Guangdong, while the other was from Rs sampled in Yunnan Among the strains from Yunnan, RsYN2 and RsYN3 were clustered together, while RsYN1 was isolated. The topology of these four strains was correlated to the sampling location. The relatively long branches reflect a high diversity among these strains, indicating a long independent evolution history.

### 3.5. Estimation of Synonymous and Nonsynonymous Substitution Rates

The *Ka/Ks* ratios (*Ks* is the number of synonymous substitutions per synonymous sites and *Ka* is the number of nonsynonymous substitutions per nonsynonymous site) were calculated for all genes. The *Ka/Ks* ratios for most of the genes were generally low, which indicates these genes were under purified selection. However, the *Ka/Ks* ratios of *ORF4a*, *ORF4b,* and *ORF9* (0.727, 0.623, and 0.843, respectively) were significantly higher than those of other ORFs (Table 4). For further selection pressure evaluation of the ORF4a and ORF4b gene, we sequenced another four ORF4a and ORF4b genes (strain Rs4223, Rs4236, Rs4240, and Ra13576 was shown in Figure 1B). The *Ka/Ks* ratios of these genes detected in 2012 (two strains) and 2013 (6 strains) were calculated, respectively. A reduction of *Ka/Ks* was observed from 2012 to 2013 (*4a*: 1.135 [2012] to 0.487 [2013]; *4b*: 4.489 [2012] o 1.764 [2013]).

### 3.6. Apoptosis Analysis of ORF9

As SARS-CoV ORF7a was reported to induce apoptosis, we conducted apoptosis analysis on BtCoV/Rh/YN2012 ORF9, a ~30% aa identity homologue of SARSr-CoV ORF7a. We transiently transfected *ORF9* of BtCoV/Rh/YN2012 into HEK293T cells to examine whether this ORF9 triggers apoptosis. Western blot was performed to confirm the expression of ORF9s and SARS-CoV ORF7a (Appendix A). ORF9 couldn’t induce apoptosis as the ORF7a of SARS-CoV Tor2 (Appendix A). The results indicated that BtCoV/Rh/YN2012 ORF9 was not involved in apoptosis induction.

### 3.7. ORF4a Proteins Induce Production of IFN-β

To determine whether these accessory proteins modulate IFN induction, we transfected reporter plasmids (pIFNβ-Luc and pRL-TK) and expression plasmids to 293T cells. All the cells over-expressing the accessory genes, as well as influenza virus NS1 (strain PR8), HBoV VP2, or empty vector were tested for luciferase activity after SeV infection. Luciferase activity stimulated by SeV was remarkably higher than that without SeV treatment as expected. Influenza virus NS1 inhibits the expression from IFN promoter, while HBoV VP2 activate the expression. Compared to those controls, the ORF4a proteins exhibit an active effect as HBoV VP2 (Figure 5A). Other accessory proteins showed no effect on IFN production (Appendix A). Expression of these accessory genes were confirmed by Western blot (Appendix A).

### 3.8. ORF3a Proteins Modulate NF-κB

NF-κB plays an important role in regulating the immune response to viral infection and is also a key factor frequently targeted by viruses for taking over the host cell. In this study, we tested if these accessory proteins could modulate NF-κB. 293T cells were co-transfected with reporter plasmids (pNF-κB-Luc and pRL-TK), as well as accessory protein-expressing plasmids, or controls (empty vector, NS1, SARS-CoV Tor2-ORF7a). The cells were mock treated or treated with TNF-α for 6 h at 24 h post-transfection. The luciferase activity was determined. RsYN1-ORF3a and RaGD-ORF3a activated NF-κB as SARS-CoV ORF7a, whereas RsYN2-ORF3a inhibited NF-κB as NS1 (Figure 5B). Expressions of ORF3as were confirmed with Western blot (Appendix A). Other accessory proteins did not modulate NF-κB production (Appendix A).

### 3.9. BtCoV/Rh/YN2012 Spike Mediated Pseudovirus Entry

To understand the infectivity of these newly detected BtCoV/Rh/YN2012, we selected the RsYN1, RsYN3 and RaGD spike proteins for spike-mediated pseudovirus entry studies. Both Western blot analysis and negative-staining electron microscopy observation confirmed the preparation of BtCoV/Rh/YN2012 successfully (Appendix A). A total of 11 human cell lines, 8 bat cells, and 9 other mammal cell lines were tested, and no strong positive was found (Appendix A).

## 4. Discussion

In this study, a novel alpha-CoV species, BtCoV/Rh/YN2012, was identified in two *Rhinolophus* species. The 4 strains with full-length genome were sequences. The 7 conserved replicase domains of these viruses possessed <90% aa sequence identity to those of other known alpha-CoVs, which defines a new species in accordance with the ICTV taxonomy standard [42]. These novel alpha-CoVs showed high genetic diversity in their structural and non-structural genes. Strain RaGD from *R. affinis*, collected in Guangdong province, formed a divergent independent branch from the other 3 strains from *R. sinicus*, sampled in Yunnan Province, indicating an independent evolution process associated with geographic isolation and host restrain. Though collected from same province, these three virus strains formed two genotypes correlated to sampling locations. These two genotypes had low genome sequence identity, especially in the S gene and accessory genes. Considering the remote geographic location of the host bat habitat, the host tropism, and the virus diversity, we suppose BtCoV/Rh/YN2012 may have spread in these two provinces with a long history of circulation in their natural reservoir, Rhinolophus bats. With the sequence evidence, we suppose that these viruses are still rapidly evolving.

Our study revealed that BtCoV/Rh/YN2012 has a unique genome structure compared to other alpha-CoVs. First, novel accessory genes, which had no homologues, were identified in the genomes. Second, multiple TRSs were found between S and E genes while other alphacoronavirus only had one TRS there. These TRSs precede *ORF3a*, *ORF3b* (only in RsYN1), and *ORF4a/b* respectively. Third, accessory gene *ORF9* showed homology with those of other known CoV species in another coronavirus genus, especially with accessory genes from SARSr-CoV.

Accessory genes are usually involved in virus-host interactions during CoV infection [43]. In most CoVs, accessory genes are dispensable for virus replication. However, an intact 3c gene of feline CoV was required for viral replication in the gut [44,45,46]. Deletion of the genus-specific genes in mouse hepatitis virus led to a reduction in virulence [47]. SARS-CoV ORF7a, which was identified to be involved in the suppression of RNA silencing [48], inhibition of cellular protein synthesis [49], cell-cycle blockage [50], and apoptosis induction [51,52]. In this study, we found that BtCoV/Rh/YN2012 ORF9 shares ~30% aa sequence identity with SARS-CoV ORF7a. Interestingly, BtCoV/Rh/YN2012 and SARSr-CoV were both detected in *R. sinicus* from the same cave. We suppose that SARS-CoV and BtCoV/Rh/YN2012 may have acquired *ORF*7a or *ORF9* from a common ancestor through genome recombination or horizontal gene transfer. Whereas, ORF9 of BtCoV/Rh/YN2012 failed to induce apoptosis or activate NF-κB production, these differences may be induced by the divergent evolution of these proteins in different pressure.

Though different BtCoV/Rh/YN2012 ORF4a share <64.4% amino acid identity, all of them could activate IFN-β. ORF3a from RsYN1 and RaGD upregulated NF-κB, but the homologue from RsYN2 downregulated NF-κB expression. These differences may be caused by amino acid sequence variations and may contribute to a viruses’ pathogenicity with a different pathway.

Though lacking of intestinal cell lines from the natural host of BtCoV/Rh/YN2012, we screened the cell tropism of their spike protein through pseudotyped retrovirus entry with human, bat and other mammalian cell lines. Most of cell lines screened were unsusceptible to BtCoV/Rh/YN2012, indicating a low risk of interspecies transmission to human and other animals. Multiple reasons may lead to failed infection of coronavirus spike-pseudotyped retrovirus system, including receptor absence in target cells, failed recognition to the receptor homologue from non-host species, maladaptation in non-host cells during the spike maturation or virus entry, or the limitation of retrovirus system in stimulating coronavirus entry. The weak infectivity of RsYN1 pseudotyped retrovirus in Huh-7 cells could be explained by the binding of spike protein to polysaccharide secreted to the surface. The assumption needs to be further confirmed by experiments.

Our long-term surveillances suggest that *Rhinolophus* bats seem to harbor a wide diversity of CoVs. Coincidently, the two highly pathogenic agents, SARS-CoV and Rh-BatCoV HKU2 both originated from *Rhinolophus* bats. Considering the diversity of CoVs carried by this bat genus and their wide geographical distribution, there may be a low risk of spillover of these viruses to other animals and humans. Long-term surveillances and pathogenesis studies will help to prevent future human and animal diseases caused by these bat CoVs.

## Figures and Tables

**Figure 1 viruses-11-00379-f001:**
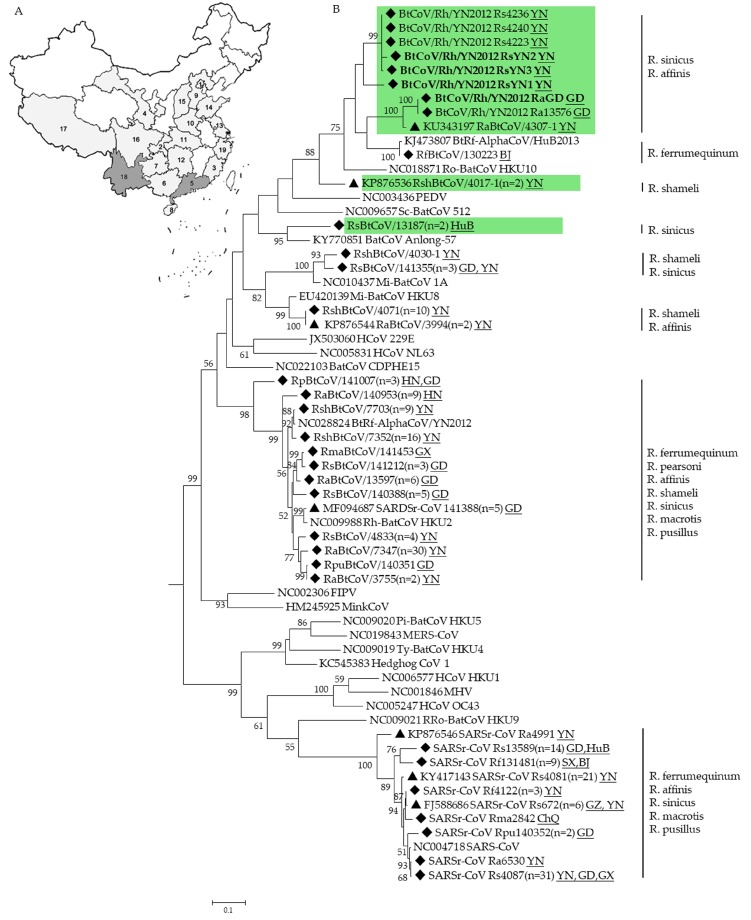
Sampling map (**A**) and phylogenetic analysis of CoVs detected in Rhinolophus bats (**B**). A total of 19 provinces (indicated in gray) in China were involved. 1. Beijing (BJ); 2. Chongqing (CA); 3. Fujian (FJ); 4. Gansu (GS); 5. Guangdong (GD); 6. Guangxi (GX); 7. Guizhou (GZ); 8. Hainan (HaN); 9. Hebei (HeB); 10. Henan (HeN); 11. Hubei (HuB); 12. Hunan (HuN); 13.Jiangsu (JS); 14.Shandong (SD); 15.Shanxi (SX); 16. Sichuan (SC). 17. Tibet (T); 18. Yunnan (YN); and 19. Zhejiang (ZJ). The partial sequences of RdRp gene (327-bp) of CoVs detected in Rhinolophus bats were aligned with those of published representative CoV strains. The tree was constructed by the maximum-likelihood method with bootstrap values determined with 1000 replicates. The scale bar indicates the estimated number of substitutions per 10 nucleotides. Filled triangles indicate the CoVs published previously by our lab (KU343197, KP876536, KP876544, MF094687, KP876546, KY417143, FJ588686) [15,18,40,41], filled diamonds indicate CoVs detected in this study. Putative novel alphaCoVs are labeled in green. BtCoV/Rh/YN2012 detected in Guangdong and Yunnan province in this study are in bold. FIPV, Feline infectious peritonitis virus; PEDV, porcine epidemic diarrhea virus; MHV, mouse hepatitis virus. Other abbreviations are defined as those in the text. Numbers in parentheses indicate numbers of sequences sharing >97% identity.

**Figure 2 viruses-11-00379-f002:**
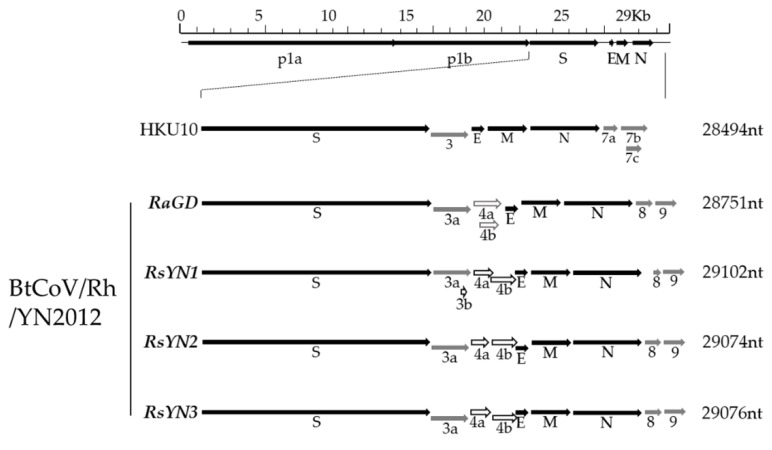
Schematic diagram of genomic organization of BtCoV/Rh/YN2012. The genomic regions or ORFs of BtCoV/Rh/YN2012 were compared with BatCoV HKU10. Solid bars indicate conserved genes and grey letters indicate species or group-specific genes. Hollow arrowheads indicate distinct array of accessory genes (Grey hollow arrowheads: RaGD; black hollow arrowheads: RsYN1, RsYN2, and RsYN3). Upper letters indicate structural proteins and lower letters indicate nonstructural proteins (p1a and p1b) and accessory proteins. HKU10, Ro-BatCoV HKU10.

**Figure 3 viruses-11-00379-f003:**
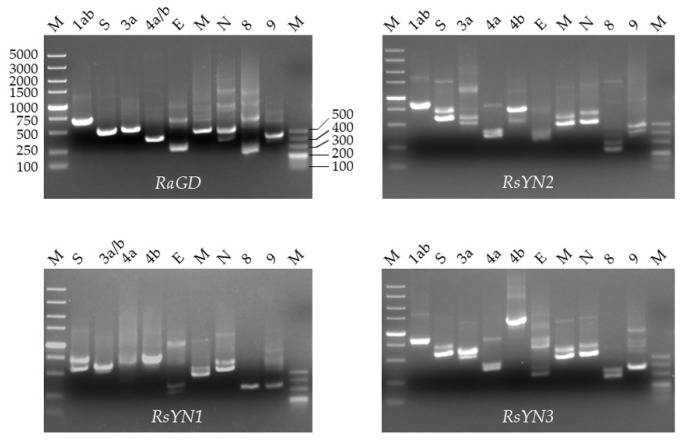
Subgenomic structure of 4 stains of BtCoV/Rh/YN2012. Agarose gel electrophoresis of the PCR products of subgenomic mRNA. The lowest band on each lane was the specific amplicon of each subgenomic mRNA. Other bands are amplicons of first-round PCR products or upper subgenomic mRNAs.

**Figure 4 viruses-11-00379-f004:**
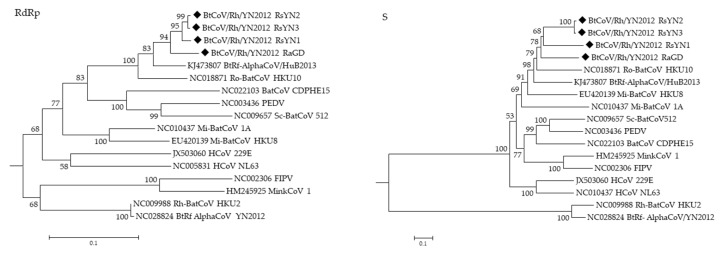
Phylogenetic trees generated based on aa sequences of RdRp and S. Trees were constructed by the maximum likelihood method using the Poisson model with bootstrap value determined with 1000 replicates. The scale bar indicates the estimated number of substitutions per 10 amino acids. Filled diamonds indicate viruses detected in this study.

**Figure 5 viruses-11-00379-f005:**
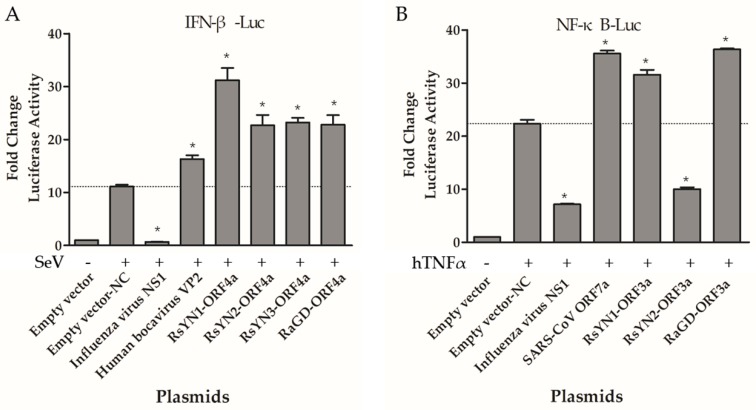
Functional analysis of BtCoV/Rh/YN2012 accessory genes. (**A**) ORF4a proteins inhibit the production of Type Ι interferon. 293T cells seeded in 24-well plates were transfected with 100 ng pIFN-β-Luc, 5 ng pRL-TK, empty vector (625 ng), an influenza A NS1-expressing plasmid (625 ng), a HBoV VP2-espressing plasmid (625 ng), or ORF4a-expressing plasmids (625 ng). The cells were infected with Sendi virus (100 hemagglutinating units/mL) at 24 h posttransfection. Samples were collected at 6 h postinfection, followed by dual-luciferase assay. The results were expressed as the firefly luciferase value normalized to that of Renilla luciferase. (**B**) ORF3a protein activate NF-κB. 293T cells were transfected with 100 ng pNF-κB-Luc, 10 ng pRL-TK, empty vector (500 ng), an NS1-expressing plasmid (500 ng), a SARS-CoV ORF7a-expressing plasmid (500 ng), or ORF3a-expressing plasmids (500 ng). After 24 h, the cells were treated with TNF-α. Dual-luciferase activity was determined after 6 h. The results were expressed as the firefly luciferase activity normalized to that of Renilla luciferase. The experiments were performed three times independently. Data are representative of at least three independent experiments, with each determination performed in triplicate (mean ± SD of fold change). Asterisks indicate significant differences between groups (compared with Empty vector-NC, *p* < 0.05, as determined by student *t* test).

**Table 1 viruses-11-00379-t001:** Detection of CoVs in *Rhinolophus* bats by RT-PCR.

Bat Species	Samples	CoV Species Positive (Positive Rate %)							Province^1^
		HKU2	HKU8	HuB2013	Mi-BtCoV 1	SARSr-CoV	BtCoV/Rh/YN2012	Other AlphaCoVs	
*R.affinis*	499	52(10.4)	2(0.4)			2(0.4)	3(0.6)		5, 7, 8, 10, 13, 18, 19
*R.blythi*	17								5,11,18
*R.ferrumequinum*	238			1(0.4)		12(5.0)			1,3,4,7,9,10,14,15,18
*R.lepidus*	21								3
*R.luctus*	8								5,8,19
*R.macrotis*	31	1(3.2)				1(3.2)			2,5,6,9,11,13,1,18
*R.monoceros*	5								18
*R.pearsoni*	106								3,5,6,7,10,17,18
*R.pusillus*	283	4(1.4)				2(0.7)			3,5,6,9,10,11,13,14
*R.shamelli*	55	25(45.5)	10(18.2)		1(0.4)			2(0.7)^2^	18
*R.sinicus*	740	12(1.6)			3(0.4)	72(9.7)	6(0.8)	2(0.3)^3^	3,5,6,7,8,11,12,13,16,18,19
*R.spp.*	58								2,6,7,18
Total	2061	94(4.6)	12(0.6)	1(0.05)	4(0.2)	89(4.3)	9(0.4)	4(0.2)	

^1^ Abbreviations for provinces: 1. Beijing; 2.Chongqing; 3. Fujian; 4. Gansu; 5. Guangdong; 6. Guangxi; 7. Guizhou; 8. Hainan; 9. Hebei; 10. Henan; 11: Hubei; 12. Hunan; 13.Jiangsu; 14.Shandong; 15.Shanxi; 16. Sichuan. 17: Tibet; 18.Yunnan; 19.Zhejiang. ^2^ Putatitive novel AlphaCoVs (RshBtCoV/4017-1), previously published.^3^ Putative novel AlphaCoVs (RsBtCoV/13187), unpublished.

**Table 2 viruses-11-00379-t002:** Comparison of coding regions, TRSs, and whole genome of BtCoV/Rh/YN2012 and with BtRf AlphaCoV/HuB2013 (HuB2013) and Ro-BatCoV HKU10 (HKU10).

CoV	ORFs	Nucleotide Positions	Predicted Size (AA) of Protein	Pairwise AA Identity (%)						TRS Sequence
				RsYN1	RsYN2	RsYN3	RaGD	HuB2013	HKU10	
RsYN1	1ab	298–20465	6722		93.0	93.0	83.7	81.8	76.4	CTAAAC(216)ATG
	S	20472–24533	1353		68.4	67.9	64.9	62.3	61.7	CTAAAT(8)ATG
	S1	20541–22634	698		60.2	59.4	56.7	53.7	53.2	
	S2	22659–24527	623		76.9	77.0	74.0	72.1	71.4	
	NS3a	24533–25201	222		65.0	65.0	54.3	52.4	53.5	CAATAC(26)ATG
	NS3b	24978–25070	30		NA	NA	NA	NA	NA	CCTTAC(35)ATG
	NS4a	25222–25581	119		47.1	62.2	18.6	NA	NA	CTTTAC(50)ATG
	NS4b	25563–26009	148		57.8	52.9	11.4	NA	NA	
	E	25993–26217	74		89.2	89.2	77.1	68.7	75.9	CTAAAC(66)ATG
	M	26229–26912	227		88.8	89.2	78.1	86.6	85.5	CTAAAC(1)ATG
	N	26923–28134	403		84.3	84.3	68.1	70.3	61.8	CTAAAC(3)ATG
	NS8	28290–28409	39		26.7	33.3	28.6	NA	NA	CTAAAC(0)ATG
	NS9	28454–28819	121		78.1	50.6	55.6	NA	NA	TTTCAC(3)ATG
	Concatenated domains^1^	13,650	4550		97.3	97.4	91.9	84.2	86.9	
RsYN2	1ab	298–20435	6712	93.0		99.4	84.4	82.5	76.9	CTAAAC(216)ATG
	S	20437–24495	1352	68.4		93.6	66.9	65.7	62.4	CTAAAT(3)ATG
	S1	20509–22599	697	60.2		88.2	58.7	58.8	55.6	
	S2	22624–24489	623	76.9		99.9	76.3	72.8	70.0	
	NS3a	24495–25154	219	65.0		100.0	55.9	52.8	55.5	CGTTAC(26)ATG
	NS4a	25164–25488	104	47.1		64.4	17.5	NA	NA	CTTCAC(24)ATG
	NS4b	25507–25953	148	57.8		94.2	12.5	NA	NA	
	E	25934–26158	74	89.2		100.0	81.9	66.3	77.1	CTAAAC(66)ATG
	M	26170–26859	229	84.3		99.8	79.9	90.0	88.8	CTAAAC(1)ATG
	N	26870–28096	408	84.3		99.8	68.3	69.9	63.0	CTAAAC(3)ATG
	NS8	28107–28385	92	26.7		66.7	64.8	NA	NA	CTAAAC(0)ATG
	NS9	28425–28790	121	78.1		77.5	58.1	NA	NA	TTTCAC(3)ATG
	Concatenated domains	13641	4547	97.3		99.6	91.8	84.3	87.1	
RsYN3	1ab	298–20435	6712	93.0	99.4		84.4	82.5	76.9	CTAAAC(216)ATG
	S	20437–24495	1352	67.9	93.6		66.6	65.8	62.4	CTAAAT(3)ATG
	S1	20506–22599	698	59.4	88.2		57.9	59.2	54.9	
	S2	22621–24489	623	77.0	99.9		76.4	72.9	70.1	
	NS3a	24495–25154	219	65.0	100.0		55.9	52.8	55.5	CGTTAC(26)ATG
	NS4a	25164–25526	120	62.2	64.4		17.6	NA	NA	CTTCAC(24)ATG
	NS4b	25508–25954	148	52.9	94.2		12.5	NA	NA	
	E	25935–26159	74	89.2	100.0		81.9	66.3	77.1	CTAAAC(66)ATG
	M	26171–26860	229	89.2	99.6		80.3	90.3	89.2	CTAAAC(1)ATG
	N	26871–28097	408	84.3	99.8		68.3	69.9	62.8	CTAAAC(3)ATG
	NS8	28108–28386	92	33.3	66.7		63.8	NA	NA	CTAAAC(0)ATG
	NS9	28434–28793	119	71.2	77.5		50.6	NA	NA	TTTCAC(3)ATG
	Concatenated domains	13641	4547	97.4	99.6		91.8	84.4	87.2	
RaGD	1ab	296–20466	6723	83.7	84.4	84.4		80.0	76.4	CTAAAC(215)ATG
	S	20463–24542	1359	64.9	66.9	66.6		61.9	62.4	CTAAAC(4)ATG
	S1	20520–22652	711	56.7	58.7	57.9		53.6	54.5	
	S2	22674–24535	621	74.0	76.3	76.4		71.4	72.2	
	NS3a	24542–25198	218	54.3	55.9	55.9		48.0	52.0	CGTTAC(26)ATG
	NS4a	25200–25663	157	18.6	17.5	17.6		NA	NA	CTTTGC(34)ATG
	NS4b	25294–25611	105	11.4	12.5	12.5		NA	NA	
	E	25654–25878	74	77.1	81.9	81.9		63.9	81.9	CCAAAC(66)ATG
	M	25884–26624	246	78.1	79.9	80.3		78.1	78.8	CTAAAC(3)ATG
	N	26635–27810	391	68.1	68.3	68.3		67.3	60.8	CTAAAC(3)ATG
	NS8	27821–28099	92	28.6	64.8	63.8		NA	NA	CTAAAC(0)ATG
	NS9	28123–28485	120	55.6	58.1	50.6		NA	NA	CTTTAC(3)ATG
	Concatenated domains	13,674	4558	91.9	91.8	91.8		83.0	87.4	

Concatenated domains including following conserved domains in replicase polyprotein pp1ab: ADRP, nsp5 (3CL^pro^), nsp12 (RdRp), nsp13 (Hel), nsp14 (ExoN), nsp15 (NendoU), and nsp16 (O-MT).

**Table 3 viruses-11-00379-t003:** Characteristics of predicted nonstructural proteins of ORF1ab in different stains of BtCoV/Rh/YN2012.

NSP	Putative Functional Domain(s)	RsYN1			RsYN2			RsYN3			RaGD		
		Amino Acides Position in ORF1ab	Predicted Size (aa of Protein)	C-end Predicted Cleavage Site	Amino Acides Position in ORF1ab	Predicted Size (aa of Protein)	C-End Predicted Cleavage Site	Amino Acides Position in ORF1ab	Predicted Size (aa of Protein)	C-end Predicted Cleavage Site	Amino Acides Position in ORF1ab	Predicted Size (aa of Protein)	C-end Predicted Cleavage Site
NSP1	Unknown	M1–A195	195	VA│AP	M1-A195	195	VA│SP	M1-A195	195	VA│SP	M1-A195	195	TA│PP
NSP2	Unknown	A196–C896	701	RC│GG	S196-S889	694	RC│GG	S196-S889	694	RC│GG	P196-S889	694	RS│GG
NSP3	ADRP, PL2 pro	G897–G2463	1567	QG│SG	G890-G2453	1564	AG│SG	G890-G2453	1564	AG│SG	G890-G2462	1573	NG│SG
NSP4	Hydrophobid domain	S2464–Q2941	478	LQ│SG	S2454–Q2931	478	LQ│SG	S2454-Q2931	478	LQ│SG	S2463-Q2940	478	LQ│SG
NSP5	3CL pro	S2942–Q3243	302	LQ│ST	S2932–Q3233	302	LQ│ST	S2932-Q3233	302	LQ│ST	S2941-Q3242	302	LQ│SN
NSP6	Hydrophobid domain	S3244–Q3519	276	VQ│SK	S3234–Q3509	276	VQ│SK	S3234-Q3509	276	VQ│SK	S3243-Q3518	276	VQ│SK
NSP7	Replicase	S3520–Q3602	83	LQ│SV	S3510–Q3592	83	LQ│SV	S3510-Q3592	83	LQ│SV	S3519-Q3601	83	LQ│SV
NSP8	Replicase	S3603–Q3797	195	LQ│NN	S3593–Q3787	195	LQ│NN	S3593-Q3787	195	LQ│NN	S3602-Q3796	195	LQ│NN
NSP9	Replicase	N3798–Q3905	108	LQ│AG	N3788–Q3895	108	LQ│AG	N3788-Q3895	108	LQ│AG	N3797-Q3904	108	LQ│AG
NSP10	RNA synthesis protein	A3906–Q4041	136	VQ│SL	A3896–Q4031	136	VQ│AL	A3896-Q4031	136	VQ│AL	A3905-Q4040	136	VQ│SL
NSP11	Unknown (short peptide at the end of ORF1a)	S–D	17		A-D	17		A-D	17		S-N	17	
NSP12	RdRp	S4042–Q4968	927	LQ│AA	A4032-Q4958	927	LQ│AA	A4032-Q4958	927	LQ│AA	S4041-Q4967	927	LQ│AA
NSP13	Hel, NTPase	A4969–Q5565	597	LQ│AG	A4959-Q5555	597	LQ│AG	A4959-Q5555	597	LQ│AG	A4968-Q5564	597	LQ│AG
NSP14	ExoN, NMT	A5566–Q6083	518	LQ│GL	A5556-Q6073	518	LQ│GL	A5556-Q6073	518	LQ│GL	A5565-Q6082	518	LQ│GL
NSP15	NeudoU	G6084–Q6422	339	LQ│AG	G6074-Q6412	339	LQ│SG	G6074-Q6412	339	LQ│SG	G6083-Q6421	339	LQ│SG
NSP16	2’-O-MT	A6423–K6722	300		S6413-K6712	300		S6413-K6712	300		S6422-K6723	302	

**Table 4 viruses-11-00379-t004:** Estimation of nonsynonymous and synonymous substitution rates in the genomes of BtCoV/Rh/YN2012.

Gene	Ka/Ks Ratio
NSP1	0.256
NSP2	0.390
NSP3	0.315
NSP4	0.276
NSP5	0.128
NSP6	0.323
NSP7	0.067
NSP8	0.125
NSP9	0.230
NSP10	0.098
NSP11	0.236
NSP12	0.073
NSP13	0.045
NSP14	0.121
NSP15	0.124
NSP16	0.083
S	0.258
ORF3a	0.357
ORF4a	0.727
ORF4b	0.623
E	0.062
M	0.0691
N	0.230
ORF8	0.276
ORF9	0.843

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
