# Peer review of "Characterization of a New Member of Alphacoronavirus with Unique Genomic Features in Rhinolophus Bats"

_viruses, 2019, doi:10.3390/v11040379_

Reviewer 1 Report

Comments on “Characterization of a New Member of Alphacoronavirus with Unique Genomic Features in Rhinolophus bats”

The manuscript by Wang et al. describes the detection of a novel alphacoronavirus with unique genomic features and accessory genes from rhinolophid bats during a 10-year lasting sample period between 2004 and 2014. Bats are known to harbor a huge diversity of viruses, many of which are zoonotic and can cause severe, life-threatening diseases in humans. Human and other non-bat alpha- and betacoronaviruses are believed to have their origin in bats and (horseshoe) bats present in South-East Asia, especially, China have been previously shown to harbor a plethora of genetically diverse coronaviruses. A prime example of a bat-borne, zoonotic virus, SARS-CoV has emerged in 2002 in China and a constant fear exist that such an spillover (due to a SARS-like CoV or another alpha- or betacoronavirus) event might happen in the future. Therefore, increased surveillance is performed to analyze coronavirus diversity in bats and identify potentially zoonotic coronaviruses before a spillover event. In this context, Wang et al. screened 2,061 bat samples from 19 Chinese provinces for the presence of coronavirus nucleic acids and found 209 of them to be positive. Without obtaining any virus isolate, the authors further performed phylogenetic analysis and identified “at least” eight different coronaviruses of which three are putative novel species. One of these novel viruses was further analyzed with respect to its genome, thereby identifying unique genome features including three additional ORFs. Besides their genomic and phylogenetic analyses the authors further performed a variety of assays to characterize some of the batAlpha-CoV proteins regarding their functional activity: They analyzed whether batAlpha-CoV ORF9, ORF4a and ORF3a have an influence on the induction of apoptosis, IFN-beta production and activation of the NF-kB pathway, respectively, using flow cytometry and luciferase-based reporter assays. Thereby, they observed that batAlpha-CoV ORF9 does not induce apoptosis (like SARS-CoV ORF7a), whereas batAlpha-CoV ORF4a and ORF3a trigger IFN-beta production and the NF-kB pathway (except for RsYN2 ORF3a), respectively. Finally, the authors tested S protein driven host cell entry into a broad panel of human, bat and other mammalian cell lines but did get clear evidence for any of the tested cell lines to be susceptible to batAlpha-CoV infection.

This study nicely connects genomic and phylogenetic analyses with tests addressing the functional activity of batAlpha-CoV proteins. The experiments are performed well, however, the functional data require expression controls (see major comments) to clarify whether variations observed between proteins of different batAlpha-CoVs (or SARS-CoV proteins that were used as control) result from differences in their expression levels. Further, readability of the manuscript would benefit from additional proof reading (see minor comments).

 Major comments:

Regarding the functional activity of batAlpha-CoV proteins, the authors performed flow cytometry and reporter assays to evaluate the effects of batAlpha-CoV ORF9, ORF4a and ORF3a on induction of apoptosis, IFN-beta production and activation of the NF-kB pathway, respectively. For comparison, the SARS-CoV ORF7a was used as an inductor of apoptosis and activator of the NF-kB pathway, whereas FLUAV-NS1 and HBoV-VP2 served as positive and negative controls for effectors of IFN-beta production. However, no expression data of the batAlpha-CoV proteins are provided which makes it sometimes difficult to interpret differences among them (IFN-beta and NF-kB assays) or with respect to SARS-CoV ORF7a (apoptosis and NF-kB assays). For example, as bat Alpha-CoV ORF9 and ORF3a are compared to SARS-CoV ORF7a with respect to induction of apoptosis and induction of the NF-kB pathway it would be interesting to compare the intracellular location of these proteins (e.g. via fluorescence microscopy). Listed below are my suggested additional experiments to increase the significance of the different reporter assays.

 (1) Figure S1: Apoptosis assay

Expression analysis of RsYN1-3 and RaGD ORF9 (and SARS-CoV ORF7a) by Western blot is needed to rule out that absence of induction of apoptosis is due to lower/insufficient expression of the different ORF9 in comparison to SARS-CoV ORF7a. If no specific/cross-reactive antibody is available to detect ORF4a, tagged versions can be used.

 (2) Figure 5: IFN-beta and Nf-kB assays

It appears that RsYN1 ORF4a induces higher levels of IFN-beta than ORF4a proteins of the other batAlpha-CoVs. Is this difference statistically significant? Is this difference due to variations in the expression levels of the ORF4a proteins (should be addressed, e.g. by Western blot)? Further, does the protein sequence of RsYN1 ORF4a markedly differ from the other ORF4a proteins (which could explain the difference)? With respect to the NF-kB assay, RsYN2 ORF3a expression does led to a decrease rather than to an increase of activation of the NF-kB pathway (as seen for the other ORF3a proteins). Again, the question remains whether this difference is statistically significant and if this difference is due to variations in the ORF3a expression levels? Also (again), does the protein sequence of RsYN2 ORF3a provide a possible explanation for the observed difference?

 (3) Table 5: Pseudotype entry studies

The “weak/positive” transduction of Huh-7 cells by pseudotypes bearing the S protein of RsYN1 would be an indicator for potential zoonotic transmission (which would be an important finding). However, it is unclear what “+/-” or “weak/positive” means, especially with respect to particles bearing MERS-CoV S or no S protein (e.g. number of positive cells, signal intensity over background). Also, the others define “+” as “positive” but particles bearing MERS-CoV S can lead to signals ranging from “-“ to “++++”. As mentioned before it is therefore important to have a clear definition what “-“, “+/-“, “+” … “++++” means.

Further, the authors state in the material and methods section that they have performed Western blot analysis and negative-staining electron microscopy to confirm pseudotype particles (but it is not clear whether the particles were analyzed or the incorporation of the different S proteins, which is more important). Those data should be included in the manuscript as they are crucial to judge the quality of S protein incorporation (especially in comparison to MERS-S). Of note, it is known for some alphacoronaviruses that their S-protein contains retention signals which reduce/block plasma membrane transport (e.g. PMID: 15304515 or 21798295). Have the batAlpha-CoV S proteins been analyzed regarding the presence of such motifs?  

 Minor comments:

General

Line 55: “gene source” does not appear to be the right term as no bat genes are integrated in the CoV genome but rather recombination of different CoV genomes leads to genetically “novel” CoVs.

 Line 57: To clearly differentiate between severe acute respiratory syndrome-related CoV (SARS-CoV) and SARS-CoV-related viruses (SARSr-CoV) it is necessary to make clear that those are not the same virus. For this, the abbreviation SARSr-CoV needs to be explained. Further, please carefully check the correct use of “SARS-CoV” and “SARSr-CoV” throughout the manuscript.

 Line 150: The reporter plasmids should be stated and their source (or a citation) should be provided.

 Line 276: I do not understand what the sentence beginning with Interestingly, when we compared […]“ means. 4a and 4b genes from 2012 to 2013 where compared, but how are the values (4a: 1.135 [2012] to 0.487 [2013]; 4b: 4.489 [2012] to 1.764 [2013]) linked to table 4?

 Line 355: Please add a reference for the statement that “BtCoV/Rh/YN2012 and SARSr-CoV were both detected in R. sinicus from the same cave”

 Line 373: The assumption that binding to polysaccharides is possible has not been tested. It should be therefore clearly stated that this is an assumption without experimental proof.

 Figure legends: The authors state that statistical significance has been tested between “groups”. To avoid confusion of the reader, the reference data set/column should be named in the figure legend or the figures should be modified in a way that it is immediately clear which data sets were compared regarding statistical significance.

 Text/Grammar

Line 25: “[…] at the downstream of nucleocapsid gene.” should be changed to “[…] downstream of the nucleocapsid gene.”

Line 40: Alphaletovirus should begin with a capital letter

Line 58: “proven” instead of “proved”

Line 89: I believe “primer sequences” and not “primers” are meant

Line 98: “manual” instead of “manually”

Line 129: “was generated” instead of “were kept”

Line 135: “centrifugation” instead of “centrifuge”

Line 135: “After incubation at 37°C for 1 h the inoculum was replaced […]” instead of “Incubate the cells at 37°C for 1h and then replaced the inoculum […]”

Line 137: “carried out” instead of “carried”

Line 138: “pellet” instead of “pallet”

Line 188: “Guangdong” instead of “Guang dong”

Line 214: There are two periods at the end of the sentence beginning with “Strain RsYN2 […]”

Line 223: “[…] of the same genotype” instead of “[…] in same genotype”

Line 226: “except for” instead of “except”

Line 233 “grey letters” instead of “greys” and “arrowheads” instead of “arrowhead”

Line 248: “lead to a” instead of “lead a”

Line 276: There needs to be a space before of “Interestingly”

Line 283: “trigger” or “increase” instead of “active”

Line 311: “modulate” instead of “regulated”

Line 338: “the S gene” instead of “S gene”

Line 347: “in another coronavirus genus” instead of “in other coronavirus genus”

Line 352, 354, 356: “SARS-CoV” instead of “SARSr-CoV”

Line 366: “were unsusceptible” instead of “was unsusceptible”

Line 375: “both originate from” instead of “are both originated”

Line 376: “genus” instead of “family”

Author Response

Major comments:

Regarding the functional activity of batAlpha-CoV proteins, the authors performed flow cytometry and reporter assays to evaluate the effects of batAlpha-CoV ORF9, ORF4a and ORF3a on induction of apoptosis, IFN-beta production and activation of the NF-kB pathway, respectively. For comparison, the SARS-CoV ORF7a was used as an inductor of apoptosis and activator of the NF-kB pathway, whereas FLUAV-NS1 and HBoV-VP2 served as positive and negative controls for effectors of IFN-beta production. However, no expression data of the batAlpha-CoV proteins are provided which makes it sometimes difficult to interpret differences among them (IFN-beta and NF-kB assays) or with respect to SARS-CoV ORF7a (apoptosis and NF-kB assays). For example, as bat Alpha-CoV ORF9 and ORF3a are compared to SARS-CoV ORF7a with respect to induction of apoptosis and induction of the NF-kB pathway it would be interesting to compare the intracellular location of these proteins (e.g. via fluorescence microscopy). Listed below are my suggested additional experiments to increase the significance of the different reporter assays.

Point 1: Figure S1: Apoptosis assay. Expression analysis of RsYN1-3 and RaGD ORF9 (and SARS-CoV ORF7a) by Western blot is needed to rule out that absence of induction of apoptosis is due to lower/insufficient expression of the different ORF9 in comparison to SARS-CoV ORF7a. If no specific/cross-reactive antibody is available to detect ORF4a, tagged versions can be used.

Response 1: We confirmed the expression of these proteins using Western blotting with an anti-HA tag monoclonal antibody (see below and Supplementary Figure S1). All target protein, except RsYN1 ORF8, was expressed. The predicted size of these proteins was described in the figure legend. The results are described in lines 307-308 and lines 319-320 in the revised manuscript. The method is described in revised 138-139.

Supplementary Figure 1: Western blot analysis of the expression of accessory proteins. The HA-tagged proteins were detected with mAb against HA tag. The bands circled in the red boxes indicate the expected proteins.

Point 2: Figure 5: IFN-beta and Nf-kB assays. It appears that RsYN1 ORF4a induces higher levels of IFN-beta than ORF4a proteins of the other batAlpha-CoVs. Is this difference statistically significant? Is this difference due to variations in the expression levels of the ORF4a proteins (should be addressed, e.g. by Western blot)? Further, does the protein sequence of RsYN1 ORF4a markedly differ from the other ORF4a proteins (which could explain the difference)? With respect to the NF-kB assay, RsYN2 ORF3a expression does led to a decrease rather than to an increase of activation of the NF-kB pathway (as seen for the other ORF3a proteins). Again, the question remains whether this difference is statistically significant and if this difference is due to variations in the ORF3a expression levels? Also (again), does the protein sequence of RsYN2 ORF3a provide a possible explanation for the observed difference?

Rsesponse 2: The difference of IFN-β and NF-κB induction by different accessory proteins is statistically different. Because the tested proteins share a low sequence identity and were in similar expression level confirmed by western blot (see above and Supplementary Figure 1), we assume that the variation of different proteins resulted in different level of induction.

Response 3: Table 5: Pseudotype entry studies. The “weak/positive” transduction of Huh-7 cells by pseudotypes bearing the S protein of RsYN1 would be an indicator for potential zoonotic transmission (which would be an important finding). However, it is unclear what “+/-” or “weak/positive” means, especially with respect to particles bearing MERS-CoV S or no S protein (e.g. number of positive cells, signal intensity over background). Also, the others define “+” as “positive” but particles bearing MERS-CoV S can lead to signals ranging from “-“ to “++++”. As mentioned before it is therefore important to have a clear definition what “-“, “+/-“, “+” … “++++” means.

Further, the authors state in the material and methods section that they have performed Western blot analysis and negative-staining electron microscopy to confirm pseudotype particles (but it is not clear whether the particles were analyzed or the incorporation of the different S proteins, which is more important). Those data should be included in the manuscript as they are crucial to judge the quality of S protein incorporation (especially in comparison to MERS-S). Of note, it is known for some alphacoronaviruses that their S-protein contains retention signals which reduce/block plasma membrane transport (e.g. PMID: 15304515 or 21798295). Have the batAlpha-CoV S proteins been analyzed regarding the presence of such motifs?

Rsesponse 3: Luciferase intensity was normalized to fold change value by that of cells infected with pseudovirus without Spike. Definition of mentioned “+/-”, “+”, and other characters are as follows. -,<2-fold increase;+/-, 2-5 fold increase; +, 5-20 fold increase; ++, 20-100 fold increase; ++++, >1000 fold increase. The definition was clarified in Supplementary Table S2.

We verified the pseudovirus production in the media supernatant using Western blotting with both anti-HIV-p24 mAb and anti-HA-tag mAb (Supplementary Figure S5A). We also collected negative-stain EM the data of these CoV-spike-pseudotyped lentiviruses. The crown-like envelope was observed (Supplementary Figure S5B). The results are described in lines 350-352 and present as Supplementary Figure S5 (also see below).

Supplementary Figure S5. Characteristics of BtCoV/Rh/YN2012 spike mediated pseudovirus. (A) Western blot analysis the protein expression of pseudovirus in the medium supernatant. (B) Electronic microscopy observation of negative-stain coronavirus particles. The bars represent 200 nm.

The YXXΦ motif (where x is any amino acid, and Φ is an aliphatic amino acid which contains a bulky hydrophobic side chain) is efficient for proteins intracellular retention. The tyrosine residue in the motif, might be part of a retention signal, was conserved. We found the peptide “YEVE” in the carboxyl terminus of these novel S proteins. The tyrosine residue is at position -9 from the carboxyl terminus. A glutamic acid (charged amino acid) is at the position -6. The glutamic acid doesn’t contain a bulky hydrophobic side chain. So we don’t think peptide “YEVE” is a YXXΦ motif.

Minor comments:

General

Point 4: Line 55: “gene source” does not appear to be the right term as no bat genes are integrated in the CoV genome but rather recombination of different CoV genomes leads to genetically “novel” CoVs.

Response 4: Thanks for your suggestion, we have re-written the sentence as “Phylogenetic analysis suggested that bats are major hosts for alpha- and beta-CoVs. Recombination of different CoVs occurred in bats as previously reported. Bats play an important role in CoV evolution.”. Please review this sentence in lines 54-56 in the revised manuscript.

Point 5: Line 57: To clearly differentiate between severe acute respiratory syndrome-related CoV (SARS-CoV) and SARS-CoV-related viruses (SARSr-CoV) it is necessary to make clear that those are not the same virus. For this, the abbreviation SARSr-CoV needs to be explained. Further, please carefully check the correct use of “SARS-CoV” and “SARSr-CoV” throughout the manuscript.

Response 5: According to ICTV proposal, human and civet SARS-CoV and bat SARS-related CoV belong to the same virus species, SARS-related coronavirus. To make the difference, we use SARS-CoV for human and bat SARSr-CoV for bats.

Point 6: Line 150: The reporter plasmids should be stated and their source (or a citation) should be provided.

Response 6: We add the reporter plasmid (pRL-TK and pIFN-β-Luc or PNF-κB-Luc) and a citation in line 158.

Point 7: Line 276: I do not understand what the sentence beginning with Interestingly, when we compared […]“ means. 4a and 4b genes from 2012 to 2013 where compared, but how are the values (4a: 1.135 [2012] to 0.487 [2013]; 4b: 4.489 [2012] to 1.764 [2013]) linked to table 4?

Response 7: We rewrote the sentence to “For further selection pressure evaluation of the ORF4a and ORF4b gene, we sequenced another four ORF4a and ORF4b (strain Rs4223, Rs4236, Rs4240 and Ra13576 as shown in Figure 1B). The Ka/Ks ratios of these genes detected in 2012 (two strains) and 2013 (6 strains) were calculated, respectively. A reduction of Ka/Ks was observed from 2012 to 2013 (4a: 1.135 [2012] to 0.487 [2013]; 4b: 4.489 [2012] o 1.764 [2013])”. (Lines 297-300 in the revised manuscript)

Point 8: Line 355: Please add a reference for the statement that “BtCoV/Rh/YN2012 and SARSr-CoV were both detected in R. sinicus from the same cave”

Response 8: This statement is based on our sampling record and the result was not published.

Point 9: Line 373: The assumption that binding to polysaccharides is possible has not been tested. It should be therefore clearly stated that this is an assumption without experimental proof.

Response 9: We appreciate the suggestion. We add “The assumption needs to be further confirmed by experiments” (line 401) to state this is a hypothesis without experimental proof.

Point 10: Figure legends: The authors state that statistical significance has been tested between “groups”. To avoid confusion of the reader, the reference data set/column should be named in the figure legend or the figures should be modified in a way that it is immediately clear which data sets were compared regarding statistical significance.

Response 10: the reference column was named Empty vector-NC. Please see the revision in Figure 5 and figure legend (revised line 337)

Figure 5. Functional analysis of BtCoV/Rh/YN2012 accessory genes. (A) ORF4a proteins inhibit the production of Type Ι interferon. 293T cells seeded in 24-well plates were transfected with 100 ng pIFN-β-Luc, 5 ng pRL-TK, empty vector (625 ng), an influenza A NS1-expressing plasmid (625 ng), a HBoV VP2-espressing plasmid (625 ng), or ORF4a-expressing plasmids (625 ng). The cells were infected with Sendi virus (100 hemagglutinating units/ml) at 24 h posttransfection. Samples were collected at 6 h postinfection, followed by dual-luciferase assay. The results were expressed as the firefly luciferase value normalized to that of Renilla luciferase. (B) ORF3a protein activate NF-κB. 293T cells were transfected with 100 ng pNF-κB-Luc, 10 ng pRL-TK, empty vector (500 ng), an NS1-expressing plasmid (500 ng), a SARS-CoV ORF7a-expressing plasmid (500 ng), or ORF3a-expressing plasmids (500 ng). After 24 h, the cells were treated with TNF-α. Dual-luciferase activity was determined after 6 h. The results were expressed as the firefly luciferase activity normalized to that of Renilla luciferase. The experiments were performed three times independently. Data are representative of at least three independent experiments, with each determination performed in triplicate (mean ± SD of fold change). Asterisks indicate significant differences between groups (compared with Empty vector -NC, P < 0.05 as determined by student t test).

 Point 11: Text/Grammar

Line 25: “[…] at the downstream of nucleocapsid gene.” should be changed to “[…] downstream of the nucleocapsid gene.”

Line 40: Alphaletovirus should begin with a capital letter

Line 58: “proven” instead of “proved”

Line 89: I believe “primer sequences” and not “primers” are meant

Line 98: “manual” instead of “manually”

Line 129: “was generated” instead of “were kept”

Line 135: “centrifugation” instead of “centrifuge”

Line 135: “After incubation at 37°C for 1 h the inoculum was replaced […]” instead of “Incubate the cells at 37°C for 1h and then replaced the inoculum […]”

Line 137: “carried out” instead of “carried”

Line 138: “pellet” instead of “pallet”

Line 188: “Guangdong” instead of “Guang dong”

Line 214: There are two periods at the end of the sentence beginning with “Strain RsYN2 […]”

Line 223: “[…] of the same genotype” instead of “[…] in same genotype”

Line 226: “except for” instead of “except”

Line 233 “grey letters” instead of “greys” and “arrowheads” instead of “arrowhead”

Line 248: “lead to a” instead of “lead a”

Line 276: There needs to be a space before of “Interestingly”

Line 283: “trigger” or “increase” instead of “active”

Line 311: “modulate” instead of “regulated”

Line 338: “the S gene” instead of “S gene”

Line 347: “in another coronavirus genus” instead of “in other coronavirus genus”

Line 352, 354, 356: “SARS-CoV” instead of “SARSr-CoV”

Line 366: “were unsusceptible” instead of “was unsusceptible”

Line 375: “both originate from” instead of “are both originated”

Line 376: “genus” instead of “family”

Response11: Thank you for your suggestion. We have made the modification accordingly.

Reviewer 2 Report

This is a review on the manuscript entitled “Characterization of a New Member of Alphacoronavirus with Unique Genomic Features in Rhinolophus bats” by Wang et al. In this manuscript, the authors describe new alphaCoV species in bats with unique genomic features. Additional ORFs encoding ORF3a, ORF4a, ORF4b and/or ORF9 were determined and functional analysis for some of these new ORFs were also performed. This manuscript is very interesting and contains valuable information to understand the biology of bat CoVs. However, the authors should clearly address the major comments raised by this reviewer before being accepted for publication in Viruses.

 Methods

Please add primer sequence information used to perform nested RT- PCRs for detection of viral subgenomic mRNAs in the CoV-positive samples as Supplementary material. Please clarify whether forward primers are targeting the leader-body junction of each sgmRNAs.

Results

Page 4, lines 170-174: It is not clear the “8 different CoVs” that is mentioned in the lines 170-171. Only seven CoV species are shown in the Table 1 including the novel alpha CoV. In line 174, “…while the other 3 viruses were putative novel CoV species”, it is not clear which novel CoV species are referred to.  Similarly, the authors claim that CoVs are deteted in 209 out of 2061 samples, but the total number of CoV detected samples in the Table 1 is 213.

Page 7, line 199: The number of positive samples for R. affinis bats in the Table 1 and in the results section (page 7, line 199) are indicated differently.

Page 7, lines 212-217: In my opinion, the text contains information about the phylogenetic tree (Figure 1B). Thus, this information should be placed accordingly. Please move either the text or the corresponding figure accordingly to facilitate reading.

Page 7, line 225-227: The aa identity of ORF9 with SARSr-CoV ORF7a is only 28.1-32% which is not enough to state that these genes are homologous. Moreover, functional analysis also showed that ORF9 does not induce apoptosis whereas SARS-CoV ORF7a does. Thus, it is possible that ORF9 does not encode protein similar to ORF7a. Could you comment on this? If the authors are no sure on the function and identity of the ORF9, the data should be removed and find a proper explanation. In addition, the authors used SARS-CoV for the apoptosis experiment instead of SARSr-CoV. Is there a reference that the SARS-CoV ORF7a induces apoptosis as well?

Page 13, lines 274-277: What are the other four ORF4a and ORF4b strains?

Page 13. Apoptosis analysis of ORF9. Please see the comment #5.

Page 15: Spike mediated pseudovirus entry and Table 5. Although the result shown in the 3.9 section and the Table 5 contain valuable information, it is not critical to demonstrate that the authors have identified novel of Bat-CoV species. Thus, the information could be moved as Supplementary material. It would be nice to include the Western blot and electron microscopy figures as supplementary material.

Discussion

Page 16, lines 349-359: Please see comment #5.

Page 16, lines 361-362: ORF4a should be changed to ORF3a. Is there any evidence on differential regulation of NF-kB among different strains of coronaviruses? To speculate that the difference of NF-kB regulation, the authors should additional data of NF-kB-Luc assay using ORFas from other isolates.

Page 16, lines 366-367: It is not correct to state that there is low risk of interspecies transmission to human and other animals because the pseudovirus particles used in this work were not susceptible to the cell lines tested here.

Overall, the manuscript has some typing and grammatical errors that should be revised carefully.

Author Response

Point 1: Please add primer sequence information used to perform nested RT- PCRs for detection of viral subgenomic mRNAs in the CoV-positive samples as Supplementary material. Please clarify whether forward primers are targeting the leader-body junction of each sgmRNAs.

Response 1: Thank you for your suggestion. We have provided the primer sequence in Supplementary Table S2. The forward primers target the leader-sequence of the genome (Line 122 in the revised manuscript).

Point 2: Page 4, lines 170-174: It is not clear the “8 different CoVs” that is mentioned in the lines 170-171. Only seven CoV species are shown in the Table 1 including the novel alpha CoV. In line 174, “…while the other 3 viruses were putative novel CoV species”, it is not clear which novel CoV species are referred to. Similarly, the authors claim that CoVs are deteted in 209 out of 2061 samples, but the total number of CoV detected samples in the Table 1 is 213.

Response : We detected 8 species of CoVs in the samples. Five of them belong to known species: SARS, HKU2, HKU8, HB2013 and MiBtCoV1. The other three are putative novel alphaCoV species: BtCoV/Rs/YN2012, RshBtCoV/4017-1, RsBtCoV/13187. Only the BtCoV/Rs/YN2012 was further analyzed. The other two novel strains were listed in Table 1 and clarified in the caption”.

We found a total of 213 CoV strains in 209 positive samples. Co-infection by two different viruses was found in 4 samples.

Point 3: Page 7, line 199: The number of positive samples for R. affinis bats in the Table 1 and in the results section (page 7, line 199) are indicated differently.

Response: We’ve corrected the text in Page 10, Line 210 in the revised manuscript. One of the three strains (RaBtCoV/4307-1) detected R.affinis has been previously reported (PMID:26920708).

Point 4: Page 7, lines 212-217: In my opinion, the text contains information about the phylogenetic tree (Figure 1B). Thus, this information should be placed accordingly. Please move either the text or the corresponding figure accordingly to facilitate reading.

Response: Thank you for your suggestion. These sentences are moved to lines 211-213.

Point 5: Page 7, line 225-227: The aa identity of ORF9 with SARSr-CoV ORF7a is only 28.1-32% which is not enough to state that these genes are homologous. Moreover, functional analysis also showed that ORF9 does not induce apoptosis whereas SARS-CoV ORF7a does. Thus, it is possible that ORF9 does not encode protein similar to ORF7a. Could you comment on this? If the authors are no sure on the function and identity of the ORF9, the data should be removed and find a proper explanation. In addition, the authors used SARS-CoV for the apoptosis experiment instead of SARSr-CoV. Is there a reference that the SARS-CoV ORF7a induces apoptosis as well?

Response: We analyzed the protein homology with HHpred software (Lines 117-118 in the revised manuscript). The result shown that ORF9s and SARS-CoV ORF7a are homologues (possibility: 100%, E value<10-48). We added this information to the results in revised line 243-245.

The reference that reported apoptosis induction by SARS-CoV ORF7: PMID 15564512. We add a citation for this reference.

Point 6: Page 13, lines 274-277: What are the other four ORF4a and ORF4b strains?

Response: The other four strains were mentioned in Figure 1B (Rs4223, Rs4236, Rs4240 and Ra13576). We add the description accordingly (Line 297 of the revised manuscript).

Page 13. Apoptosis analysis of ORF9. Please see the comment #5.

Point 7: Page 15: Spike mediated pseudovirus entry and Table 5. Although the result shown in the 3.9 section and the Table 5 contain valuable information, it is not critical to demonstrate that the authors have identified novel of Bat-CoV species. Thus, the information could be moved as Supplementary material. It would be nice to include the Western blot and electron microscopy figures as supplementary material.

Response: Thank you for your suggestion. Table 5 was moved as Supplementary Table S3. We also added the Western blot and electron microscopy figures as Supplementary Figure S5.

Supplementary Figure 5. Characteristic of BtCoV/Rh/YN2012 spike mediated pseudovirus. (A) Western blot analysis the protein expression of pseudovirus in the medium supernatant. (B) Electronic microscopy observation of negative-stain coronavirus particles. The bars represent 200 nm.

 Discussion

Page 16, lines 349-359: Please see comment #5.

Point 8: Page 16, lines 361-362: ORF4a should be changed to ORF3a. Is there any evidence on differential regulation of NF-kB among different strains of coronaviruses? To speculate that the difference of NF-kB regulation, the authors should additional data of NF-kB-Luc assay using ORFas from other isolates.

Response: We used “ORF3a” to replace “ORF4a” in the revised manuscript (line 389 in the revises manuscript). Previous publications lack the evidence that differential regulation of NF-κB among different CoVs strains. Bat SARS-related CoV ORF3b homologues display different interferon antagonist activities have been reported.

The RsYN2 ORF3a shared 100% aa identity to RsYN3 ORF3a. We suppose both of them down-regulated NF-κB. Besides RsYN1, RsYN2 and RaGD, no additional genotype of ORF3a was detected in this study.

Point 9: Page 16, lines 366-367: It is not correct to state that there is low risk of interspecies transmission to human and other animals because the pseudovirus particles used in this work were not susceptible to the cell lines tested here.

Response: The pseudoviruses entry into Huh-7 cells are presented in the following figure. We use MERS-CoV-S-mediated pseudovirus, no spike (mock), and the pseudovirus entry to HEK293T as controls. The controls worked as previously reported (PMID: 23532101 & 234806063). The related luciferase unit counts of BtCoV/Rh/YN2012 RsYN1 S pseudovirus was about 4-fold higher than mock. The P-value was lesser than 0.05. Therefore, we state that there may be a low risk of spillover. (line 405 in the revised manuscript)

Figure. BtCoV/Rs/YN2012-S and MERS-CoV-S-mediated pseudovirus entry. Error bat indicate SEM, * P<0.05.< span="">

Point 10: Overall, the manuscript has some typing and grammatical errors that should be revised carefully.

Response: Thank you for your suggestion. We have checked these errors through the text.

Reviewer 3 Report

In this study the authors present several new fragmentary coronavirus sequences and four ostensibly complete genomes.  If the abstract is taken at face value, there are also an extensive number of biological tests to investigate protein function and tropism.

I think these observations are fundamentally publishable.  The biological tests plus the sequence data would justify publication in this journal if documented more thoroughly.  I think the presentation needs revision, though.

There are several experiments that the authors mention, but do not show.  If the experiments have been done and are presentable, then this would be a minor revision, but if some of the controls have not been done, then it will be a more extensive revision.

Larger concerns:

1. line 28 and Figure 5 - Virus proteins can be difficult to work with, and sometimes differences in the outcome of a test like this can potentially be attributed to differences in expression level.  The reader needs more information in order to evaluate this, including expression controls.  Do you have a gel that shows that proteins of the expected size were expressed from these plasmids?  Even something like a Transcription and Translation kit could help show that the genes are at least in principle expressible from these constructs.  The experimental setup is OK otherwise, I think, and I like the idea to include this test.

2. The manuscript (correctly) cites the current coronavirus taxonomy, which comes from the ICTV Master Species List 2018b, but cites references 2 and 3.  Both are old, reflect a different and more primitive taxonomy, and neither is a particularly good choice to justify current taxonomy - the alphaletovirus mentioned in the introduction, for example, wasn't published until 2018.  Also line 53 implies that references 11 and 12 are ICTV publications, which I don't believe is correct - change these statements or change the references please.

3. line 92 - what kind of deep sequencing?  Please talk about the method, how the reads were assembled, any important variations (such as leader-body junctions), coverage for each genome, and some kind of quality statistic for the run.

4. line 161 - please show EM results for the new virus pseudotyped VLPs and the MERS-CoV control.  Coronavirus VLPs are readily identifiable by their long spikes, and since this is the only control that would demonstrate that VLPs were produced and spikes were incorporated, it is important.  Otherwise, we would need to see protein expression for S, and evidence that S proteins were incorporated into VLPs.  If these are like some other alphacoronaviruses, the spikes may be a little longer than the MERS-CoV spikes, and it would be nice to evaluate whether that is the case.

5. line 223 - Please tighten up the wording - the protein encoded by ORF3a may be identical for each virus, but the ORF itself cannot be, or else they would all have an identical ORF3b too.

6. Figure 2 - Please indicate in some way that the 4a and 4b proteins of RaGD are not homologous to the 4a and 4b of the RsYN1-3 viruses, or demonstrate that they are.  A good test would be to use something like hhpred to compare the RaGD and RsYN versions of the protein.

7. Table 5 - Please show pictures to document the Huh-7 result (the only one that may have shown entry, for one kind of pseudotyped particle).

8. Lines 175-6 and abstract - please make it clear that virus isolation was attempted, but was not successful.  As written it took me a while to figure that out.

Minor concerns:

25 - found at the downstream -- > found downstream

25-26 - by RT-PCR and full genome sequencing?

85 - reference or list the primer sequences please.  It's OK that primers in section 2.4 (below) are not mentioned since they are totally specific to this virus, but the primers mentioned here are of general use and would be of wider interest.

127 - by RT-PCR surely?

138 - pallet should be pellet

Table 2 - Please remove the line "genome" - it is nucleic acid, and cannot have amino acid identity. The concatenated domains below make the point that these viruses are closely related, at least in the key domains of the replicase.

Table 3 and line 101 - Please change putative cleavage sites to predicted cleavage sites, and discuss the basis for the prediction in the results or discussion section in more detail - it looks like this may have been done manually, which can be OK if the proteins being compared are good homologs, but should be explicitly stated.

Table 3 - presumably nsp7-10 are homologous and equivalent in function to SARS-CoV nsp7-10?  An hhpred or even BLAST match with good statistics would help justify this.

Author Response

 Comments and Suggestions for Authors

In this study the authors present several new fragmentary coronavirus sequences and four ostensibly complete genomes. If the abstract is taken at face value, there are also an extensive number of biological tests to investigate protein function and tropism.

I think these observations are fundamentally publishable. The biological tests plus the sequence data would justify publication in this journal if documented more thoroughly. I think the presentation needs revision, though.

There are several experiments that the authors mention, but do not show. If the experiments have been done and are presentable, then this would be a minor revision, but if some of the controls have not been done, then it will be a more extensive revision.

Larger concerns:

Point 1: line 28 and Figure 5 - Virus proteins can be difficult to work with, and sometimes differences in the outcome of a test like this can potentially be attributed to differences in expression level. The reader needs more information in order to evaluate this, including expression controls. Do you have a gel that shows that proteins of the expected size were expressed from these plasmids? Even something like a Transcription and Translation kit could help show that the genes are at least in principle expressible from these constructs. The experimental setup is OK otherwise, I think, and I like the idea to include this test.

Response: We appreciated your suggestions. We verified the expression of these proteins using Western blotting with an anti-HA tag monoclonal antibody (Supplementary Figure S1). Except RsYN1 ORF8, we detected target protein expression as expected. The predicted size of these proteins were described in the figure legend. The results are described in lines 307-308 and lines 319-320 in the revised manuscript. The method is described in revised lines 138-139.

Supplementary Figure 1: Western blot analysis of the expression of accessory proteins. The HA-tagged proteins were detected with mAb against HA tag. The bands circled in the red boxes indicate the expected proteins.

Point 2: The manuscript (correctly) cites the current coronavirus taxonomy, which comes from the ICTV Master Species List 2018b, but cites references 2 and 3.  Both are old, reflect a different and more primitive taxonomy, and neither is a particularly good choice to justify current taxonomy - the alphaletovirus mentioned in the introduction, for example, wasn't published until 2018.Also line 53 implies that references 11 and 12 are ICTV publications, which I don't believe is correct - change these statements or change the references please.

Response 2: Thank you for your suggestion. We changed the reference to ICTV Master Species List 2018b in the revised manuscript. (Revised line 41and line 53)

Point 3: line 92 - what kind of deep sequencing? Please talk about the method, how the reads were assembled, any important variations (such as leader-body junctions), coverage for each genome, and some kind of quality statistic for the run.

Response 3: In this study, we first amplified the genomic fragments by long RT-PCR with primers designed from alphacoronavirus sequence alignment and/or specific sequences. PCR products over 5kb were subjected to Next Generation Sequencing (NGS) using the Hiseq 2500 system. Original NGS data were filtered and mapped to the reference sequence of Ro-BatCoV HKU10 (GenBank accession number NC_018871) using Geneious 7.1.8. This method was briefly described in lines 94-95 and lines 98-99 in the revised manuscript.

We were not able to detect the leader-body junctions from the NGS data.

Point 4: line 161 - please show EM results for the new virus pseudotyped VLPs and the MERS-CoV control. Coronavirus VLPs are readily identifiable by their long spikes, and since this is the only control that would demonstrate that VLPs were produced and spikes were incorporated, it is important. Otherwise, we would need to see protein expression for S, and evidence that S proteins were incorporated into VLPs. If these are like some other alphacoronaviruses, the spikes may be a little longer than the MERS-CoV spikes, and it would be nice to evaluate whether that is the case.

Response 4: Thank you for the suggestion. We verified the pseudovirus production in the media supernatant using Western blotting with both anti-HIV-p24 mAb and anti-HA-tag mAb (Supplementary Figure S5A). We also collected negative-stain EM the data of these CoV-spike-pseudotyped lentiviruses. The crown-like envelope was observed (Supplementary Figure S5B). However, we don’t think the resolution of these negative-stain EM photos could support comparison among the length of these spikes. The results are described in revised lines 350-352.

Supplementary Figure S5. Characteristics of BtCoV/Rh/YN2012 spike mediated pseudovirus. (A) Western blot analysis the protein expression of pseudovirus in the medium supernatant. (B) Electronic microscopy observation of negative-stained coronavirus particles. The bars represent 200nm.

Point 5: line 223 - Please tighten up the wording - the protein encoded by ORF3a may be identical for each virus, but the ORF itself cannot be, or else they would all have an identical ORF3b too.

Response 5: we using “the proteins encoded by ORF3a” instead of “the ORF3a” (Lines 239-240 in the revised manuscript)

Point 6: Figure 2 - Please indicate in some way that the 4a and 4b proteins of RaGD are not homologous to the 4a and 4b of the RsYN1-3 viruses, or demonstrate that they are. A good test would be to use something like hhpred to compare the RaGD and RsYN versions of the protein.

Response 6: We analyzed the protein homology with HHpred software. We described this method in lines 117-118 in the revised manuscript. We revised Figure 2. Grey hollow arrowheads indicate RaGD ORF4a and ORF4b, and black hollow arrowheads indicate RsYN1-3 ORF4a and ORF4b. The description was shown in lines 253-254 in the revised manuscript.

Figure 2. Schematic diagram of genomic organization of BtCoV/Rh/YN2012. The genomic regions or ORFs of BtCoV/Rh/YN2012 were compared with BatCoV HKU10. Solid bars indicate conserved gene and grey letters indicate species or group-specific genes. Hollow arrowheads indicate distinct array of accessory genes (Grey hollow arrowheads: RaGD; black hollow arrowheads: RsYN1, RsYN2 and RsYN3). Upper letters indicate structural proteins and lower letters indicate nonstructural proteins (p1a and p1b) and accessory proteins. HKU10, Ro-BatCoV HKU10.

Point 7: Table 5 - Please show pictures to document the Huh-7 result (the only one that may have shown entry, for one kind of pseudotyped particle).

Response 7: The entry into Huh-7 cells are presented in the following figure. We used MERS-CoV-S-mediated pseudovirus, no spike (NC), and the pseudovirus entry to HEK293T as controls. The controls worked as previously reported (PMID:23532101 & 234806063). The related luciferase unit counts of BtCoV/Rs/YN2012 S pseudovirus was about 4-fold higher than NC. The P-value was lesser than 0.05.

Figure. BtCoV/Rs/YN2012-S and MERS-CoV-S-mediated pseudovirus entry. Error bat indicate SEM, * P<0.05.< span="">

Point 8: Lines 175-6 and abstract - please make it clear that virus isolation was attempted, but was not successful. As written it took me a while to figure that out.

Response 8: “and no cytopathic effect or viral replication was detected” has been replaced by “but was not successful” (Line 184 in the revised manuscript)

Minor concerns:

Point 9: 25 - found at the downstream -- > found downstream

Response 9: we using “found downstream” instead of “found at the downstream” in the revised line 25.

Point 10: 25-26 - by RT-PCR and full genome sequencing?

Response 10: The putative gene sequences were acquired by full-length genome sequencing, and then we confirmed the sequences by RT-PCR with specific primers to eliminate the possibility of false amplification of DNA polymerase. Here, we use “further confirmed” instead of confirmed” (Line 26 in the revised manuscript).

Point 11: 85 - reference or list the primer sequences please. It's OK that primers in section 2.4 (below) are not mentioned since they are totally specific to this virus, but the primers mentioned here are of general use and would be of wider interest.

Response 11: The degenerate primer sequences were listed in Supplementary Table S1 and described in the revised manuscript (revised line 88).

Point 12: 127 - by RT-PCR surely?

Response 12: We using “RT-PCR” instead of “PCR” (revised line 133).

Point 13: 138 - pallet should be pellet

Response 13: “pallet” was changed to “pellet” (revised line 145)

Point 14: Table 2 - Please remove the line "genome" - it is nucleic acid, and cannot have amino acid identity. The concatenated domains below make the point that these viruses are closely related, at least in the key domains of the replicase.

Response 14: We appreciated your suggestion. The comparison of the genomes in table 2 was removed.

Point 15: Table 3 and line 101 - Please change putative cleavage sites to predicted cleavage sites, and discuss the basis for the prediction in the results or discussion section in more detail - it looks like this may have been done manually, which can be OK if the proteins being compared are good homologs, but should be explicitly stated.

Response 15:putative” was changed to “predicted” (revised line 268 and table 3).

The replicase gene, ORF1ab, occupies ~20.4kb of the genome. It encodes polyproteins 1a and 1ab which could be cleavage into 16 non-structural proteins (Nsp1-Nsp16). The polyprotein 1a and 1ab was aligned with 16 non-structural proteins of other alphaCoV. The 3’-end of the cleavage sites could be recognized and cleaved by 3C-like proteinase (Nsp4-Nsp10, Nsp12-Nsp16) and papain-like proteinase (Nsp1-Nsp3) were confirmed. The data have been presented in lines 220-225 in the revised manuscript.

Point 16: Table 3 - presumably nsp7-10 are homologous and equivalent in function to SARS-CoV nsp7-10? An hhpred or even BLAST match with good statistics would help justify this.

Response 16: We appreciate your suggestion. The nsp7-nsp10 have been identified as co-factors of CoVs replicase in several CoVs. Nsp7-nsp10 of BtCoV/Rh/YN2012 shared<54.0% aa sequence identity to SARS-CoV nsp7-10, and they are homologous (homology detection with HHpred, possibility =100%). The functions of nsp7-10 were shown in second column of Table 3.

Reviewer 4 Report

Wang et al. describe the characterization of novel alphacoronaviruses from Rhinolophus bats. The authors obtained full-length genomes of novel bat coronavirus (BatCoV) and detected ORF4a/b gene from these viruses that have no homologues in GenBank database. The study provides useful information about the epidemiology of BatCoV but much more work is needed to obtain more consistent data.

 Major criticism:

The authors should clarify the GenBank accession number of four novel BatCoVs (RsYN1, RsYN2, RsYN3, and RaGD strains).

 line 174 and Figure 1: Please highlight BatCoVs that the author have identified as "novel CoV species" in Figure 1. In addition, please describe the reason judged as “novel CoV species” in the result or the discussion section.

 Why did the authors speculate that the 4a gene of BatCoVs induces IFN-beta or induces/deduces NF-kB? Did the authors check other genes or proteins? To begin with, did these viruses propagate in bats? (I guess that the data in Table 5 reflects the inability to propagate in bats…) Without confirming this fact, it would be useless data to measure antiviral factors such as IFN.

 It is necessary to describe how to calculate Ka / Ks ratio. Especially 4a / b gene. Could the authors calculate Ka / Ks ratio with genes that are not present in other alpha coronaviruses?

 lines 280-285: Why not do this experiment with bat-derived cells? Based on previous report, human-derived cells were used in the experiment using SARS-CoV (virus infected with human). On the basis of this fact, the authors should use bat-derived cells.  

 Minor concerns:

Abbreviations should be formed from the first letters of the word or phrase.

-SARSr-CoV (line 57)

 lines 39-41: The latest classific outline of Coronaviridae (including Letovirinae) is not on the 9th report of the ICTV (2012) and Woo’s paper (2009). Please correct references.

 lines 170-171: The authors described “Partial RdRp sequences suggested the presence of at least 8 different CoVs.” However, only seven CoVs are listed in Table 2. Did “Novel AlphaCoV” contain two species of CoVs? If so, please specify these viruses.

 Lines 349-350: This sentence is incorrect. The authors should refer to the latest report.

Deletion of the ORF 3c gene increases the virulence of feline coronaviruses (FCoVs). Deletion of ORF 7b eliminates the virulence of FCoVs.

Author Response

Comments and Suggestions for Authors

Wang et al. describe the characterization of novel alphacoronaviruses from Rhinolophus bats. The authors obtained full-length genomes of novel bat coronavirus (BatCoV) and detected ORF4a/b gene from these viruses that have no homologues in GenBank database. The study provides useful information about the epidemiology of BatCoV but much more work is needed to obtain more consistent data.

Major criticism:

Point 1: The authors should clarify the GenBank accession number of four novel BatCoVs (RsYN1, RsYN2, RsYN3, and RaGD strains).

Response 1: We appreciate your suggestion. We modified the sentence as “the complete genome nucleotide sequences of BtCoV/Rh/YN2012 strains RsYN1, RsYN2, RsYN3 and RaGD obtained in this study have been submitted to GenBank under MG9169001 to MG9169004. (lines 171-172 in the revised manuscript)

Point 2: line 174 and Figure 1: Please highlight BatCoVs that the author have identified as "novel CoV species" in Figure 1. In addition, please describe the reason judged as “novel CoV species” in the result or the discussion section.

Response 2: The novel alphaCoVs are in grey in revised Figure 1B. BtCoV/Rh/YN2012, RshBtCoV/4017-1 and RsBtCoV 13187 shared<83% nt identities to known CoVs. We assume these viruses are new species. BtCoV/Rh/YN2012 is confirmed to be a novel alphaCoV species. But currently we still do not have sufficient data to strongly support that RshBtCoV/4017-1 and RsBtCoV 13187 are novel CoV species.

We have made a revision. We using “While the other three CoV sequences showed less than 83% nt identity to known CoV species. These three viruses should represent novel CoV species” (Lines 182-183 in the revised manuscript)

Point 3: Why did the authors speculate that the 4a gene of BatCoVs induces IFN-beta or induces/deduces NF-kB? Did the authors check other genes or proteins? To begin with, did these viruses propagate in bats? (I guess that the data in Table 5 reflects the inability to propagate in bats…) Without confirming this fact, it would be useless data to measure antiviral factors such as IFN.

Response 3: Usually the accessory proteins of coronaviruses are antagonists of host immune response. In this study, we conducted a preliminary investigation of the function of all accessory proteins.

These viruses propagate in bats for following reasons. Firstly, BtCoV/Rh/YN2012 was detected at different years and different locations, indicating this virus circulating in bats. Secondly, viruses’ transcription in bats was confirmed by subgenomic RNAs sequencing. Indicating this virus could infect, replicate and transcript in host cells. These S-mediated pseudoviruses lack the ability to enter RsKT, RsLu4323, RsBrT, and RaK4324 may result from the limitation of retrovirus system (PMID: 26552008) and/or the absence of receptor expression on the cell surface.

Point 4: It is necessary to describe how to calculate Ka / Ks ratio. Especially 4a / b gene. Could the authors calculate Ka / Ks ratio with genes that are not present in other alpha coronaviruses?

Response 4: We use Ka/Ks calculation tool, Norwegian Bioinforatics Platform (http://services.cbu.uib.no/tools/kaks) to calculate the Ka/Ks ratio. The description was added to revised lines 116-117. The calculation of Ka/Ks needs at least three homologue sequences using the server. We only calculate the intraspecies Ka/Ks ratio among these viruses. No other alphacoronavirus sequence was needed.

Point 5: lines 280-285: Why not do this experiment with bat-derived cells? Based on previous report, human-derived cells were used in the experiment using SARS-CoV (virus infected with human). On the basis of this fact, the authors should use bat-derived cells.

Response 5: We appreciate your suggestion. First, it’s hard to transfect the plasmid DNA to bat-derived cells. Second, human cells can be used for evaluating the potential interspecies transmission of bat viruses.

Point 6: Abbreviations should be formed from the first letters of the word or phrase.

-SARSr-CoV (line 57)

Response 6: The abbreviation of SARSr-CoVs were explained with SARS-related coronaviruses (revised lines 58-59).

Point 7: lines 39-41: The latest classific outline of Coronaviridae (including Letovirinae) is not on the 9th report of the ICTV (2012) and Woo’s paper (2009). Please correct references.

Response 7: Thank you for your suggestion. We changed the reference to ICTV Master Species List 2018b in the revised manuscript. (Revised line 41 and line 53)

Point 8: lines 170-171: The authors described “Partial RdRp sequences suggested the presence of at least 8 different CoVs.” However, only seven CoVs are listed in Table 2. Did “Novel AlphaCoV” contain two species of CoVs? If so, please specify these viruses.

Response 8: In fact, we detected 8 species of CoVs in this paper. Five of them belong to known species: SARS, HKU2, HKU8, HB2013 and MiBtCoV1. The other three of them are putative novel alphaCoV species: BtCoV/Rs/YN2012, RshBtCoV/4017-1, RsBtCoV/13187. Only BtCoV/Rs/YN2012 was further studied (Lines 204-205 in the revised manuscript). The information of these two CoVs was presented in modified Table 1.

Point 9: Lines 349-350: This sentence is incorrect. The authors should refer to the latest report.

Deletion of the ORF 3c gene increases the virulence of feline coronaviruses (FCoVs). Deletion of ORF 7b eliminates the virulence of F CoVs.

Response 9: Thank you for pointing out the inaccurate citation. We have made the revision. “Accessory genes involved in virus-host interactions during CoV infection. In most CoVs, accessory genes are dispensable for virus replication. However, an intact 3c gene of feline CoV was required for viral replication in the gut.” The description was added in revised lines 376-378.

Round  2

Reviewer 1 Report

The authors have addressed all my questions/remarks. In particular, the authors included additional data (i.e. expression controls for ORF3a, ORF3b, ORF4a, ORF4b, ORF8, ORF9; expression controls and pseudotype incorporation data for the different S proteins), clarified the pseudotype data (transduction efficiency is now clearly defined by the reporter gene activity and compared to empty pseudotypes) and modified the text to clarify statements/conclusions and to improve the readability.

 I have only a couple minor points regarding English language and data presentation that should be addressed to make this manuscript (in my view) suitable for acceptance.

 Minor points:

 Data presentation

Figure 1, panel B: The gray shading to highlight the suspected novel alphaCoVs is hard to see. I would be good to use a darker shading of a different way to highlight the respective CoVs.

 Text/Grammar

Line 29: Change “[…] using spike-pseudotype retroviruses system, […]” to “[…] using a retroviral spike-pseudotype system, […]”

Line 31: Change “[…] will help us further […]” to “[…] will help us to further […]”

Line 134: Change “[…] were generated […]” to “[…] was generated […]”

Line 149: Delete the “by” in “[…] using by the Annexin V-FITC/PI Apoptosis Detection Kit […]”

Line 157: Change “pNF-κN-Luc” to “pNF-κB-Luc”

Line 171: Delete the “the” in “[…] submitted to the GenBank […]”

Line 210: Change “[…] bat,and 6 R.sinicus respectively.” to “[…] bats,_and 6 R.sinicus bats respectively.”

Line 211: Change “[…] three genotypes which represented by […]” to “[…] three genotypes which are represented by […]”

Line 219-226: After modification of the manuscript it seems like some of the old text has been incompletely deleted (please check and modify accordingly). In my opinion the following adjustments are needed: First, delete the first of the duplicated sentences reading “The replicase gene, ORF1ab, occupies ~20.4kb of the genome.” [Line 219]. Second, delete or modify the sentence “The proteins including Nsp3 (papain-like 2 protease, PL2pro), Nsp5 (chymotrypsin-like protease, 3CLpro), Nsp12 (RdRp), Nsp13 (helicase) and other proteins of unknown function (Table 3).” [Line 224-226].

Line 220: Change “[…] which could be cleavage into […]” to “[…] which can be cleaved into […]”

Line 221: Change “The polyprotein 1a and 1b was aligned with […]” to “The polyproteins 1a and 1b were aligned with […]”

Line 223: Change “The 3’-end of the cleavage sites could be recognized and cleaved by 3C-like proteinase (Nsp4-Nsp10, Nsp12-Nsp16) and papain-like proteinase (Nsp1-Nsp3) were confirmed.” to “The 3’-end of the cleavage sites for proteolytic processing by the 3C-like proteinase (Nsp4-Nsp10, Nsp12-Nsp16) and the papain-like proteinase (Nsp1-Nsp3) were identified.”

Line 242: Change “The results shown […]” to “The results showed […]”

Line 248 (legend to Figure 2): Change “[…] conserved gene […]” to “[…] conserved genes […]”

Line 278: Add a period after “Yunnan”

Line 308: Change “ORF4a proteins active production of IFN-β” to “ORF4a proteins increase production of IFN-β”

Line 312: Change “[…] after SeV infected.” to “[…] after SeV infection.”

Line 314: Change “activate” to activates

Line 350-351: I would consider rewriting the paragraph describing the pseudotype data. My suggestion would be something like: “A total of 11 human cell lines, 8 bat cells and 9 other mammalian cell lines were tested. None of these cell lines was clearly positive for susceptibility to transduction mediated by the S proteins of RsYN1, RsYN3 and RaGD (Only for Huh-7 cells a slight increase in reporter activity was observed following incubation with particles pseudotyped with the S protein of RsYN1) (Supplementary Table S2).

Line 369: Change “alphacoronavirus” to “alphacoronaviruses

Line 373: There is a typo in “usually

Line 376: Delete the “which” before “[…] was identified to […]”

Line 386: Change “activates” to “activated

Line 387: Change “These differences may be caused by amino acid sequences and […]” to “These differences may be caused by amino acid sequence variations between ORF3a proteins and […]”

Reference list:

Reference #42 is incorrectly displayed (names of the editors).

Author Response

Response to Reviewers’ comments

Reviewer 1

The authors have addressed all my questions/remarks. In particular, the authors included additional data (i.e. expression controls for ORF3a, ORF3b, ORF4a, ORF4b, ORF8, ORF9; expression controls and pseudotype incorporation data for the different S proteins), clarified the pseudotype data (transduction efficiency is now clearly defined by the reporter gene activity and compared to empty pseudotypes) and modified the text to clarify statements/conclusions and to improve the readability.

I have only a couple minor points regarding English language and data presentation that should be addressed to make this manuscript (in my view) suitable for acceptance.

Minor points:

Data presentation

Point 1: Figure 1, panel B: The gray shading to highlight the suspected novel alphaCoVs is hard to see. I would be good to use a darker shading of a different way to highlight the respective CoVs.

Response 1: We appreciate your suggestion. We use green shading to highlight the suspected novel alphaCoVs (Revised Figure 1 and revised line 196).

Figure 1. Sampling map (A) and phylogenetic analysis of CoVs detected in Rhinolophus bats (B). A total of 19 provinces (indicated in gray) in China were involved. 1. Beijing (BJ); 2.Chongqing (CA); 3. Fujian (FJ); 4. Gansu (GS); 5. Guangdong (GD); 6. Guangxi (GX); 7. Guizhou (GZ); 8. Hainan (HaN); 9. Hebei (HeB); 10. Henan (HeN); 11. Hubei (HuB); 12. Hunan (HuN); 13.Jiangsu (JS); 14.Shandong (SD); 15.Shanxi (SX); 16. Sichuan (SC). 17. Tibet (T); 18. Yunnan (YN); 19. Zhejiang (ZJ). The partial sequences of RdRp gene (327-bp) of CoVs detected in Rhinolophus bats were aligned with those of published representative CoV strains. The tree was constructed by the maximum-likelihood method with bootstrap values determined with 1000 replicates. The scale bar indicates the estimated number of substitutions per 10 nucleotides. Filled triangles indicate the CoVs published previously by our lab (KU343197, KP876536, KP876544, MF094687, KP876546, KY417143, FJ588686) [15, 18, 40, 41], filled diamonds indicate CoVs detected in this study. Suspected novel alphaCoVs are labeled in green. BtCoV/Rh/YN2012 detected in Guangdong and Yunnan province in this study are in bold. FIPV, Feline infectious peritonitis virus; PEDV, porcine epidemic diarrhea virus; MHV, mouse hepatitis virus. Other abbreviations are defined as those in the text. Numbers in parentheses indicate numbers of sequences sharing >97% identity

Text/Grammar

Point 2:

Line 29: Change “[…] using spike-pseudotype retroviruses system, […]” to “[…] using a retroviral spike-pseudotype system, […]”

Line 31: Change “[…] will help us further […]” to “[…] will help us to further […]”

Line 134: Change “[…] were generated […]” to “[…] was generated […]”

Line 149: Delete the “by” in “[…] using by the Annexin V-FITC/PI Apoptosis Detection Kit […]”

Line 157: Change “pNF-κN-Luc” to “pNF-κB-Luc”

Line 171: Delete the “the” in “[…] submitted to the GenBank […]”

Line 210: Change “[…] bat,and 6 R.sinicus respectively.” to “[…] bats,_and 6 R.sinicus bats respectively.”

Line 211: Change “[…] three genotypes which represented by […]” to “[…] three genotypes which are represented by […]”

Line 219-226: After modification of the manuscript it seems like some of the old text has been incompletely deleted (please check and modify accordingly). In my opinion the following adjustments are needed: First, delete the first of the duplicated sentences reading “The replicase gene, ORF1ab, occupies ~20.4kb of the genome.” [Line 219]. Second, delete or modify the sentence “The proteins including Nsp3 (papain-like 2 protease, PL2pro), Nsp5 (chymotrypsin-like protease, 3CLpro), Nsp12 (RdRp), Nsp13 (helicase) and other proteins of unknown function (Table 3).” [Line 224-226].

Line 220: Change “[…] which could be cleavage into […]” to “[…] which can be cleaved into […]”

Line 221: Change “The polyprotein 1a and 1b was aligned with […]” to “The polyproteins 1a and 1b were aligned with […]”

Line 223: Change “The 3’-end of the cleavage sites could be recognized and cleaved by 3C-like proteinase (Nsp4-Nsp10, Nsp12-Nsp16) and papain-like proteinase (Nsp1-Nsp3) were confirmed.” to “The 3’-end of the cleavage sites for proteolytic processing by the 3C-like proteinase (Nsp4-Nsp10, Nsp12-Nsp16) and the papain-like proteinase (Nsp1-Nsp3) were identified.”

Line 242: Change “The results shown […]” to “The results showed […]”

Line 248 (legend to Figure 2): Change “[…] conserved gene […]” to “[…] conserved genes […]”

Line 278: Add a period after “Yunnan”

Line 308: Change “ORF4a proteins active production of IFN-β” to “ORF4a proteins increase production of IFN-β”

Line 312: Change “[…] after SeV infected.” to “[…] after SeV infection.”

Line 314: Change “activate” to activates”

Line 350-351: I would consider rewriting the paragraph describing the pseudotype data. My suggestion would be something like: “A total of 11 human cell lines, 8 bat cells and 9 other mammalian cell lines were tested. None of these cell lines was clearly positive for susceptibility to transduction mediated by the S proteins of RsYN1, RsYN3 and RaGD (Only for Huh-7 cells a slight increase in reporter activity was observed following incubation with particles pseudotyped with the S protein of RsYN1) (Supplementary Table S2).”

Line 369: Change “alphacoronavirus” to “alphacoronaviruses”

Line 373: There is a typo in “usually”

Line 376: Delete the “which” before “[…] was identified to […]”

Line 386: Change “activates” to “activated”

Line 387: Change “These differences may be caused by amino acid sequences and […]” to “These differences may be caused by amino acid sequence variations between ORF3a proteins and […]”

Response 2: Thank you for your suggestion. We have made the modification accordingly.

Reference list:

Point 3: Reference #42 is incorrectly displayed (names of the editors).

Response 3: We correct the reference #42 in lines 557-561 in the revised manuscript.

Reviewer 2 Report

The authors addressed this reviewer's comments accordingly. I do not have further comments.

Author Response

Point: The authors addressed this reviewer's comments accordingly. I do not have further comments.

Response: Thank you for your decision.

Reviewer 3 Report

The revision addresses the concerns that I had quite thoroughly.  With the revised data and additional supplementary data, I think this is in a publishable form.

Author Response

Point: The revision addresses the concerns that I had quite thoroughly. With the revised data and additional supplementary data, I think this is in a publishable form.

Response: Thank you for your decision.

 Reviewer 4 Report

The manuscript has been much improved and is in a nice condition now.

Author Response

Point: The manuscript has been much improved and is in a nice condition now.

Response: Thank you for your decision.